# Evolving Quantitative Reasoning through Self-Play in Digital Twin Markets

**Tianmi Ma**[1]  **Wenxin Huang**[2]  **Jiawei Du**[3 4]  **Lin Li**[1]  **Xian Zhong**[1]  **Joey Tianyi Zhou**[3 4]

## Abstract

Large language models (LLMs) demonstrate strong capabilities in high-level semantic reasoning and strategic planning, making them appealing for complex decision-making tasks; however, their quantitative reasoning remains unreliable despite recent progress in tool-augmented and structured inference. To address this limitation, we **decouple reasoning from computation** by assigning LLMs to planning, analysis, and result interpretation, while delegating numerical computation and statistical inference to specialized external tools. Rather than being hard-coded, these tools are constructed in a constrained and structured manner during planning as **explicit intermediate reasoning artifacts**, enabling adaptive and scenario-dependent quantitative reasoning. LLMs iteratively analyze tool outputs under diverse market conditions and leverage performance-based feedback to refine subsequent tool selection and construction, thereby **forming a bounded self-evolving loop**. We instantiate this process through **self-play in a controllable digital twin market, DECOUPLEDMARKET**, where LLM agents continuously test, compare, and adapt their strategies. By coupling high-level planning with robust quantitative execution, the proposed framework enhances the quantitative reliability of LLM-driven decision-making. All code and data are available at https://github.com/MTMQuantAI/Agent-Trading-Arena.git.

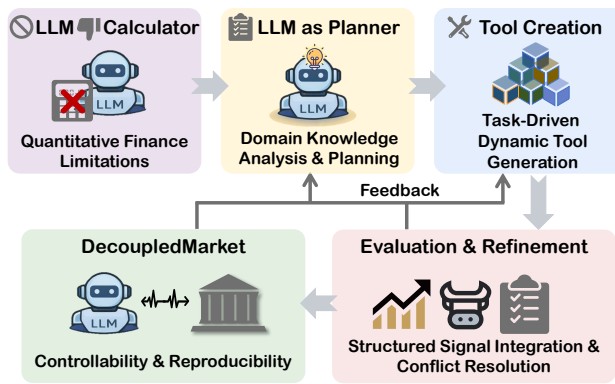

*Figure 1.* **Planning-driven quantitative tool creation and evaluation framework.** LLMs act as high-level planners that analyze market conditions, select analytical strategies, and generate task-specific quantitative tools. Structured feedback from the controllable digital twin market DECOUPLEDMARKET supports iterative evaluation and refinement, closing the loop from semantic reasoning to quantitative execution.

## 1. Introduction

Large language models (LLMs) excel at high-level semantic reasoning (Wu et al., 2024a), abstraction (Lee et al., 2025; Singhal & Shroff, 2025), and strategic planning (Sel et al., 2025; Du et al., 2025; Park et al., 2023), making them appealing for complex decision-support tasks over heterogeneous information sources. Yet, although LLMs perform well on formal mathematical problems (Kostikova et al., 2026; Zhang & Luo, 2025; Hu et al., 2024), such symbolic reasoning differs fundamentally from decision-critical quantitative tasks. In these settings, numerical errors can propagate under uncertainty, particularly in domains characterized by strong nonlinearity, compounding effects, or threshold-based rules. This limitation is especially pronounced in financial analysis, where even minor inaccuracies may translate into substantial strategic risk.

This gap between high-level semantic reasoning and **robust quantitative decision-making under uncertainty** (Ahn et al., 2024) constrains the use of LLMs as standalone quantitative decision-makers. Rather than forcing a monolithic model to internalize both semantic understanding and precise computation, we adopt a decoupled perspective: LLMs

---

[1]Hubei Key Laboratory of Transportation Internet of Things, School of Computer Science and Artificial Intelligence, Wuhan University of Technology, China [2]Hubei Key Laboratory of Big Data Intelligent Analysis and Application, School of Computer Science, Hubei University, China [3]Centre for Frontier AI Research, Agency for Science, Technology and Research (A*STAR), Singapore [4]Institute of High Performance Computing, Agency for Science, Technology and Research (A*STAR), Singapore. Correspondence to: Jiawei Du <dujw@a-star.edu.sg>, Xian Zhong <zhongx@whut.edu.cn>.

*Proceedings of the 43rd International Conference on Machine Learning*, Seoul, South Korea. PMLR 306, 2026. Copyright 2026 by the author(s).

are responsible for domain-knowledge-driven analysis and planning, while specialized tools perform precise numerical computation (Yuan et al., 2025) and statistical inference under the guidance of high-level semantic and interpretive reasoning.

Building on this perspective, we propose a **planning-driven tool-creation paradigm for quantitative reasoning**, as illustrated in Figure 1. In contrast to prior approaches (Haque et al., 2025; Wu et al., 2024b) that rely on fixed sets of predefined analytical tools, our framework enables LLMs to dynamically create task-specific quantitative tools during the reasoning process. These tools may take the form of novel indicators, feature transformations, or data-driven diagnostic procedures. Rather than being hard-coded, they are endogenously generated, structured, and orchestrated as explicit intermediate reasoning artifacts, enabling adaptive, scenario-specific integration of high-level planning with concrete quantitative computation.

Within this paradigm, LLMs engage in an iterative, evaluation-driven refinement process. They analyze the outputs of newly generated tools across diverse market scenarios and risk profiles, and use performance-based feedback, such as stability, sensitivity, and robustness, to guide subsequent tool creation, adaptation, and recomposition. This interaction forms a bounded, self-refining reasoning loop: exploration in the tool space is constrained by predefined execution budgets and evaluation criteria, while still permitting progressive analytical evolution within controlled environments (Liu et al., 2026). Refinement is driven by structured diagnostic signals rather than explicit reward optimization or adversarial self-play.

Evaluating such adaptive behaviors directly in real financial markets is costly and confounded by delayed and entangled feedback, limiting causal attribution. To address this challenge, we introduce **DECOUPLEDMARKET, a digital twin market** that provides a low-cost, reproducible, and fully controlled environment for assessing tool performance, stability, and interaction effects under explicitly specified endogenous dynamics. By decoupling evaluation from real-market execution, DECOUPLEDMARKET enables rigorous and fine-grained analysis of adaptive quantitative reasoning while preserving fidelity to essential market mechanisms.

Within DECOUPLEDMARKET, we implement an LLM-driven trading analysis agent, `DeMAgent`, as an auxiliary decision-support module. Rather than executing trades, `DeMAgent` focuses on analysis planning, quantitative tool construction and composition, and structured interpretation of computational outputs, while final decisions remain with human users or downstream systems. This design allows the agent to iteratively expand its analytical repertoire within decoupled digital twin markets.

Finally, we examine the transferability of reasoning behaviors learned in digital twin markets to real-world financial data. Agents that exhibit stronger structured reasoning and numerical abstraction achieve more stable performance on historical datasets. While this does not imply causal equivalence, the results suggest that DECOUPLEDMARKET serves as an effective surrogate for evaluating tool-creation-based reasoning and transferable structural properties of quantitative analysis beyond static backtesting.

In summary, our contributions are threefold:

- We formalize a tool-creation-based quantitative reasoning paradigm that positions LLMs as planners rather than calculators, showing that separating semantic planning from numerical execution improves the reliability of decision-critical quantitative analysis;

- We introduce DECOUPLEDMARKET, a decoupled digital twin market with endogenous price formation, enabling low-cost, controllable, and fine-grained evaluation of created quantitative tools beyond replay-based backtesting;

- We demonstrate that structured reasoning and tool abstraction developed in virtual markets can generalize to real financial data, suggesting that digital twin self-play provides a practical testbed for evaluating adaptive quantitative reasoning.

## 2. Related Work

### 2.1. LLM-Based Agents for Quantitative Reasoning

Recent studies enhance LLM-based agents with external computation modules or executable code to improve numerical reliability (Liu et al., 2025; Mirzadeh et al., 2025; Zhang et al., 2024b). While these approaches alleviate certain numerical errors, planning, tool invocation, and numerical execution are typically entangled within a single reasoning loop. This tight coupling often relegates quantitative computation to a secondary role, limiting transparency, error attribution, and systematic analysis of stability. In contrast, our framework explicitly decouples semantic reasoning from numerical computation, positioning LLMs as high-level planners and treating numerical components as independently evaluable reasoning artifacts.

### 2.2. Tool Use and Tool Creation in LLM-Based Agents

To address the quantitative limitations of LLMs, prior work equips agents with external computational tools (Wölflein et al., 2025; Rietschel & Steinfeld, 2025; Lin & Xu, 2025). Most approaches rely on predefined tool sets, while more recent efforts explore autonomous code generation or task-specific tool synthesis. However, generated tools are often

treated as transient execution artifacts, and existing studies rarely provide controlled analyses that isolate the effects of tool quality or iterative refinement. By contrast, our framework elevates quantitative tools to explicit intermediate reasoning artifacts that are created, evaluated, and iteratively adapted within a bounded loop, enabling clearer attribution between semantic planning, numerical execution, and downstream outcomes.

## 2.3. Evaluation of Financial Agents and Market Simulators

Existing evaluations of LLM-based financial agents predominantly rely on historical backtesting or static benchmarks (Gao et al., 2023; Wu et al., 2023; Nie et al., 2023), in which market prices are treated as exogenous and agent actions do not influence market dynamics. Even multi-agent market simulators (Li et al., 2026; Xiao et al., 2024; Chen et al., 2025; Zhang et al., 2024a; Li et al., 2024; Cheng & Chin, 2024) typically assume externally specified price processes, limiting their ability to reveal causal relations between agent reasoning and observed outcomes.

By contrast, our framework employs a controllable environment with explicit endogenous market dynamics, allowing agent actions to directly shape market evolution. This setting shifts evaluation toward the structure and adaptation of quantitative reasoning, including tool creation and refinement, rather than trading performance alone.

## 3. Proposed Method

We formulate this setting as a synthetic form of self-play, where an LLM-based agent iteratively refines its decision-making and tool use through repeated interactions with a dynamic, non-stationary market populated by heuristic or fixed-strategy participants. A formal definition of the self-play loop is provided in Appendix A. Figure 2 illustrates the overall architecture of the proposed framework, which comprises two core components: (1) DECOUPLEDMARKET, a digital twin market with endogenous price formation driven by agent interactions, and (2) a heterogeneous population of trading agents operating under a unified market protocol. By eliminating reliance on replayed historical price trajectories, the framework establishes a closed-loop market in which prices emerge directly from agent behavior, ensuring that observed performance reflects intrinsic reasoning rather than exogenous signals.

## 3.1. DECOUPLEDMARKET: A Digital Twin Market for Multi-Agent Trading

Evaluating trading agents in real markets is challenging due to non-stationarity, delayed feedback, and information leakage in historical data, which obscure whether observed performance arises from genuine quantitative reasoning or spurious correlations. To address these challenges, we introduce DECOUPLEDMARKET, a fully isolated digital twin market designed for controlled and competitive evaluation.

**Decoupling Strategy Evaluation from Market Noise.** DECOUPLEDMARKET isolates agents' decision-making from uncontrollable market realizations by fixing the market protocol while allowing prices, liquidity, and volatility to emerge endogenously from agent interactions. The environment does not replay historical price trajectories, expose absolute time or calendar information, or define future prices in advance; all prices are realized solely through trading actions. By controlling unreproducible exogenous shocks, historical artifacts, and implicit information leakage, which are particularly salient for LLMs with broad financial priors, this design enables fair, reproducible, and causally grounded evaluation of trading strategies (see Figure 3).

**Endogenous Multi-Agent Market Dynamics.** DECOUPLEDMARKET is a self-evolving simulation in which prices are formed endogenously from collective agent behavior. Heterogeneous agents, including classical technical strategies, standalone LLMs, and LLM-based agents, interact through a centralized matching engine under a unified protocol. Latent market states are generated online from these interactions rather than replayed from historical data. Consequently, observed performance differences are primarily attributable to strategy design, while the environment supports adaptive learning, strategic competition, and emergent phenomena such as feedback loops, crowding, and adversarial dynamics that are inaccessible in static simulators.

**Endogenous Non-Stationarity under Partial Observability.** Trading in DECOUPLEDMARKET is modeled as a partially observable Markov game in which agents infer the evolving behavior of others from aggregated and delayed market signals. Formally, the market consists of an agent set $\mathcal{I} = \{1, \ldots, M\}$. Each agent $i$ follows a policy $\pi_i(a_t^i \mid o_t^i)$, while the latent global state $s_t \in \mathcal{S}$ evolves according to:

$$\mathcal{T}\left(s_{t+1} \mid s_t, \mathbf{a}_t\right), \quad \mathbf{a}_t = \left(a_t^1, \ldots, a_t^M\right), \quad (1)$$

where transitions are jointly induced by collective agent actions and the centralized matching engine. Although the environment is Markovian with respect to $s_t$, continual adaptation of other agents' policies induces effective non-stationarity from the perspective of any individual agent $i$, yielding an induced transition:

$$\mathcal{T}_i^{\pi_{-i}}\left(s_{t+1} \mid s_t, a_t^i\right), \quad (2)$$

which marginalizes over the actions of other agents generated by their time-varying policies $\pi_{-i}$. Each agent thus operates in a non-stationary decision process induced by a

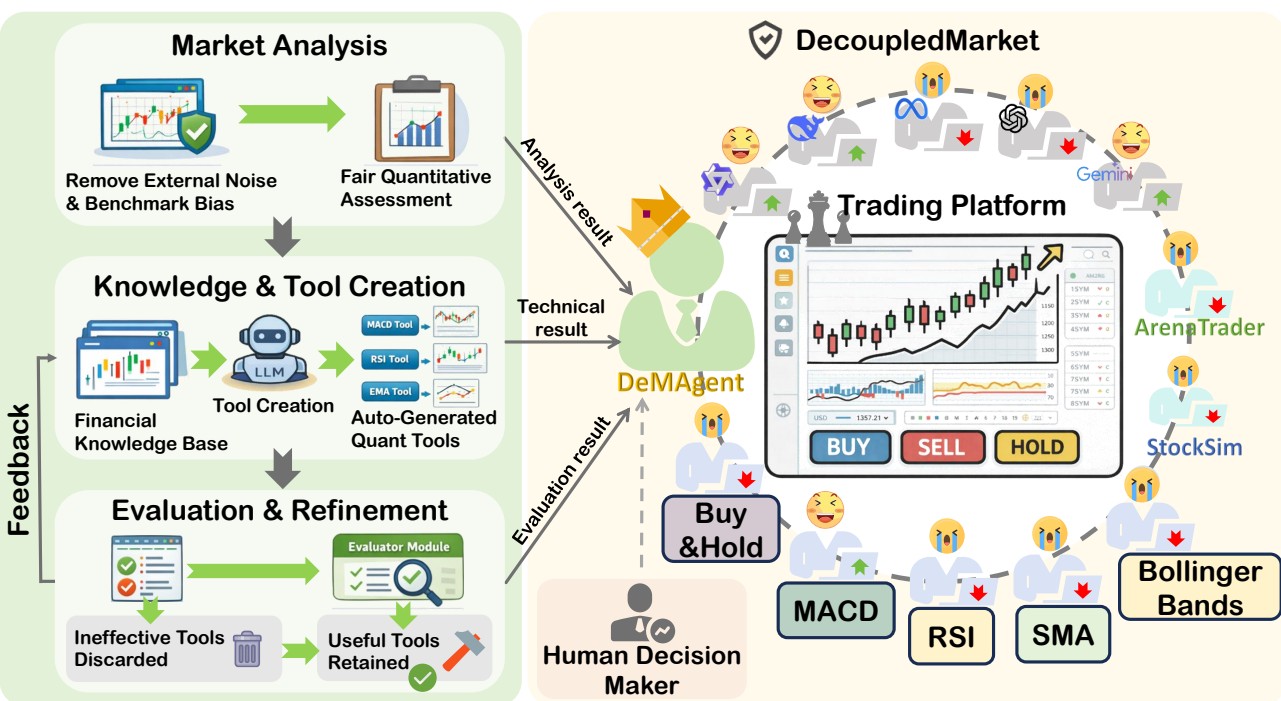

*Figure 2.* **DECOUPLEDMARKET and DeMAgent architecture.** Heterogeneous agents, including classical technical strategies, standalone LLMs, and LLM-based agents, interact through a simulated exchange in DECOUPLEDMARKET. Prices emerge endogenously from trading actions, and the resulting market states are fed back to agents to form a closed-loop market.

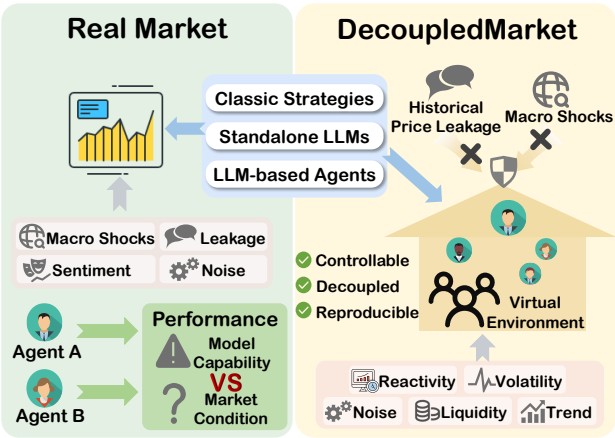

*Figure 3.* **Decoupling environmental variability from strategy evaluation.** Exogenous noise and uncontrolled variability in real markets obscure the relation between strategy design and observed performance. DECOUPLEDMARKET controls exogenous factors while preserving essential market microstructure, enabling fair, reproducible, and causally grounded evaluation.

Markov game with fixed environment dynamics but adaptively changing agent policies. We formalize DECOUPLED-MARKET as the tuple:

$$(O, \mathcal{S}, A, \mathcal{T}, Z, R, \gamma), \tag{3}$$

where $O$ and $A$ denote the agent observation and action spaces, $Z$ the agent-indexed observation function, $R$ the agent-specific trading rewards, and $\gamma$ the discount factor. At time $t$, agent $i$ receives an observation $o_{t-D+1:t}^i$, a length-$D$ window of partial market signals, where $D$ is an agent-side design choice. Observations over $N$ tradable stocks are given by $o_t^i = Z(s_t, i)$. Actions $a_t^i \in \mathbb{R}^N$ represent portfolio allocation decisions, and rewards are defined by realized trades and mark-to-market portfolio returns.

**Endogenous Price Formation.** In DECOUPLEDMAR-KET, asset prices evolve endogenously from agent interactions rather than being replayed from historical trajectories. At each timestep, agents submit portfolio-level allocation actions, which are mapped by a centralized matching engine to executed buy and sell quantities. Prices arise from supply-demand imbalances generated by executed trades within each timestep. Let $\Delta_n^t$ denote the excess demand of stock $n$ at timestep $t$:

$$\Delta_n^t = \sum_{i \in \mathcal{I}} q_{n,i}^{\text{buy},t} - \sum_{i \in \mathcal{I}} q_{n,i}^{\text{sell},t}, \tag{4}$$

where $q_{n,i}^{\text{buy},t}$ and $q_{n,i}^{\text{sell},t}$ are the quantities executed at a common reference price $\tilde{P}_{\text{close},n}^t$. To account for liquidity con-

ditions, price impact is scaled by recent trading activity:

$$\text{Impact}_n^t = \kappa \cdot \tanh\left(\frac{\Delta_n^t}{L_n^t}\right), \quad L_n^t = \sum_{\tau=t-T}^{t} V_n^\tau, \quad (5)$$

where $V_n^\tau$ is the executed volume of stock $n$ at timestep $\tau$, $L_n^t$ summarizes recent market activity rather than ex ante order-book depth, and $\kappa$ is chosen to ensure bounded impacts. The resulting transaction price is:

$$P_{\text{trade},n}^t = \tilde{P}_{\text{close},n}^t \cdot \left(1 + \text{Impact}_n^t\right). \quad (6)$$

Rather than modeling full high-dimensional microstructure, we summarize short-horizon price formation using a compact set of representative statistics. The reference price for agent valuation is defined as:

$$P_n^{t+1} = f\left(P_{\text{trade},n}^t, P_{\text{vwap},n}^t, P_{\text{trend},n}^t\right), \quad (7)$$

where $P_{\text{vwap},n}^t$ reflects volume-weighted transactional consensus, $P_{\text{trend},n}^t$ captures short-horizon price pressure induced by persistent order flow, and $f(\cdot)$ is a deterministic aggregation function designed to preserve scale consistency. Additional microstructural signals can be incorporated within this framework (see Appendix B for details).

### 3.2. `DeMAgent`: LLM-Driven Modular Trading Agent with Tool Creation

`DeMAgent` is a modular LLM-based trading agent operating within DECOUPLEDMARKET under endogenous price formation and strategic competition. For each decision, it executes a structured three-stage reasoning pipeline and leverages self-play feedback to progressively refine its reasoning and tool usage.

**Market Analysis.** The market analysis stage aggregates multi-level information, including macro trends, volatility regimes, sector- and asset-level signals, and agent-induced order-flow statistics. It distills heterogeneous observations into concise, structured abstractions that characterize the prevailing market context. These abstractions do not directly trigger actions; instead, they ground subsequent stages, allowing the agent to determine *what* should be computed before specifying *how*.

**Knowledge & Tool Creation.** This stage converts high-level market understanding into executable quantitative behavior. Conditioned on market analysis and embedded knowledge priors, the LLM synthesizes task-specific tools (*e.g.*, technical indicators or feature extractors) that serve as the sole interface for numerical computation. Tool creation is inherently constrained: generated tools must be syntactically valid, numerically sound, causally correct, and operationally safe. Even minor violations, such as look-ahead

bias or temporal misalignment, can invalidate evaluation while remaining difficult to detect. As a result, tool synthesis constitutes a constrained program generation problem with a sharply restricted valid space.

Formally, given the observation history $o_{t-D+1:t}^i$, the LLM produces a tool set $\mathcal{G}$ such that:

$$\phi_t^i = \mathcal{G}\left(o_{t-D+1:t}^i\right), \quad (8)$$

subject to hard constraints on causality, execution safety, and interface compliance. Constraint violations result in execution failure rather than soft performance degradation.

These challenges are addressed through prompt conditioning, operation masking, and interface-constrained delegation. All numerical computations are performed by the generated tools, with free-text calculation disallowed. This design converts LLM reasoning into verifiable artifacts and cleanly separates strategic decision-making from low-level computation, enabling faithful and reproducible evaluation.

**Evaluation & Refinement.** In the final stage, the LLM aggregates outputs from the generated tools and complementary signals, including regime analysis, valuation anchoring, and technical indicators, to produce a trading decision executed in DECOUPLEDMARKET. Evaluation focuses on tool effectiveness rather than the trading action itself. Through prompt-based self-assessment designed with task-specific financial criteria and a structured multi-dimensional format, the LLM considers prior reasoning, tool outputs, and observed market responses to assess whether the tools yield stable, consistent, and informative signals. The resulting feedback is propagated to upstream modules: the evaluation report identifies underperforming tool components, and the knowledge & tool creation module uses this diagnostic signal to modify existing tools or synthesize new ones, forming a closed-loop refinement cycle across trading rounds without updating model parameters.

This pipeline allows `DeMAgent` to guide high-level decisions while delegating all numerical computation and statistical estimation to tools, ensuring interpretability, stability, and a clear separation between planning and execution.

## 4. Experimental Results

### 4.1. Experimental Setup

**Environment.** We evaluate our framework in DECOUPLEDMARKET, an interactive multi-agent trading environment with endogenous price formation and leakage-free evaluation. Prices emerge from agent interactions, enabling the study of strategic behavior without relying on historical trajectories. The market contains $N$ stocks (see Table 1) and $M$ heterogeneous agents, including classical technical strategies, standalone LLMs, and LLM-based agents. Each

*Table 1.* **Stock statistics.** $P_0$ denotes the initial closing price. $\mu$ and $\sigma$ denote the historical mean and standard deviation, respectively, characterizing each stock's return distribution. DPS denotes dividend per share, and Initial Qty denotes the initial number of shares.

| ID | Ticker | $P_0$ | $\mu$ | $\sigma$ | DPS | Initial Qty |
|----|--------|-------|-------|----------|-----|-------------|
| 1 | A | 486.55 | 0.0024 | 0.0188 | 2 | 2,800 |
| 2 | B | 535.30 | 0.0142 | 0.0362 | 3 | 2,500 |
| 3 | C | 355.75 | -0.0118 | 0.0132 | 5 | 3,000 |
| ⋮ | ⋮ | ⋮ | ⋮ | ⋮ | ⋮ | ⋮ |

agent maintains holdings, cash, and trade history, and all interactions are mediated by a centralized matching engine under a unified market protocol. Stocks are specified by initial price, historical volatility, and sector, and all agents are initialized with identical capital. Stock prices evolve endogenously through trading activity. Detailed configurations are provided in Appendices C and D.

**Real-World Data.** For real-world evaluation, we consider three market benchmarks: NASDAQ, CSI, and CRYPTOCURRENCY. The evaluation periods are 1 September-26 October 2025, 1 October-30 November 2025, and 1 August-31 October 2025, respectively. These benchmarks span distinct market characteristics, including a highly volatile cryptocurrency market, a moderately trending U.S. equity market driven by earnings-related events, and the Chinese stock market with distinct trading patterns and regulatory constraints. All simulations exclude weekends and official holidays. Market data are obtained from YAHOO FINANCE and BAOSTOCK, including daily OHLC prices, trading volumes, and derived technical indicators.

**Evaluation Metrics.** Agents are initialized with role-specific capital, and we report: (1) **Total Return:** $\text{TR} = \frac{C_1 - C_0}{C_0}$, where $C_0$ and $C_1$ are the initial and final asset values. (2) **Win Rate:** $\text{WR} = \frac{N_w}{N_t}$, where $N_w$ is the number of winning trades and $N_t$ is the total number of trades. (3) **Sharpe Ratio:** $\text{SR} = \frac{(R_p - R_f)}{\sigma_p}$, where $R_p$ is the mean daily return, $\sigma_p$ is the standard deviation of daily returns, and $R_f$ is set to 0. (4) **Maximum Drawdown:** $\text{MD} = \max_{0 \le t \le T} \frac{M_t - V_t}{M_t}$, where $V_t$ is the portfolio value and $M_t = \max_{0 \le s \le t} V_s$ is the running maximum. (5) **Mean Daily Return $R_p$:** the average daily return over the trading period. (6) **Standard Deviation of Daily Returns $\sigma_p$:** daily return volatility.

**Baselines.** We compare DeMAgent against classical technical strategies (Buy & Hold, ATR, RSI, ZMR, Bollinger Bands, SMA, and MACD), reinforcement learning methods (PPO, A2C, TD3, and DDPG), standalone LLM traders (GPT-3.5-Turbo, GPT-4.1-mini, GPT-5-mini, LLaMa-3.1,

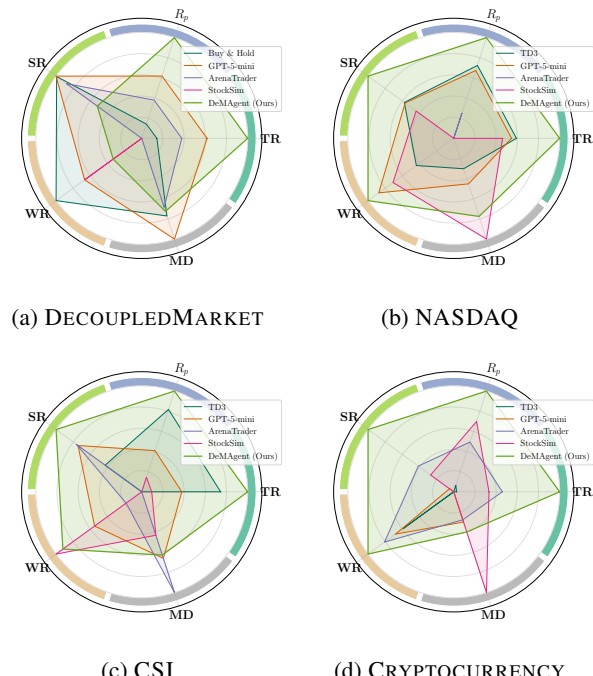

(a) DECOUPLEDMARKET  (b) NASDAQ

(c) CSI  (d) CRYPTOCURRENCY

*Figure 4.* **Cross-market performance comparison in digital twin and real markets.** Across DECOUPLEDMARKET, NASDAQ, CSI, and CRYPTOCURRENCY, DeMAgent generally achieves stronger returns and risk control than classical strategies, standalone LLMs, and prior LLM-agent baselines.

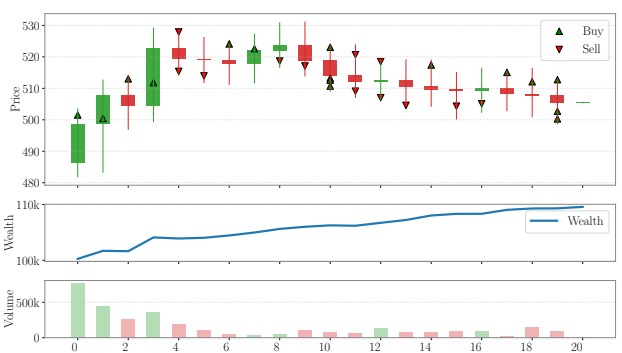

*Figure 5.* **Trading behavior and portfolio evolution for Stock A in DECOUPLEDMARKET.** Candlestick prices are overlaid with buy and sell actions generated by DeMAgent w/ GPT-5-mini, together with portfolio wealth and trading volume, illustrating how sequential decisions translate into executed trades and cumulative performance.

Qwen-3, DeepSeek-V3.2, and Gemini-3-Pro), and LLM-based agent frameworks (StockSim and ArenaTrader). More details are provided in Appendix E.

### 4.2. Overall Trading Performance

**Multi-Agent, Leakage-Free Simulation in DECOUPLED-MARKET.** We first evaluate DeMAgent in DECOUPLED-

*Table 2.* **Backbone-wise trading performance on DECOUPLEDMARKET.** Comparison of classical technical strategies, standalone LLMs, LLM-based trading agents, and DeMAgent under a unified digital twin market. DeMAgent achieves strong returns and competitive risk-adjusted performance across LLM backbones.

| Model | TR ↑ | $R_p$ ↑ | $\sigma_p$ ↓ | SR ↑ | WR ↑ | MD ↑ |
|---|---|---|---|---|---|---|
| Buy & Hold | 4.9998 | 0.2445 | 0.2340 | 1.0449 | 95.00 | -0.1561 |
| ATR (Wilder, 1978) | 1.8331 | 0.0909 | 0.1313 | 0.6925 | 80.00 | -0.1750 |
| RSI (Wilder, 1978) | 4.8543 | 0.2375 | 0.2216 | 1.0716 | 85.00 | -0.2265 |
| ZMR (Poterba & Summers, 1988) | 0.9212 | 0.0459 | 0.0854 | 0.5375 | 35.00 | -0.1013 |
| Bollinger Bands (Bollinger, 1992) | 0.3616 | 0.0181 | 0.0283 | 0.6382 | 35.00 | **0.0000** |
| SMA (Gencay, 1996) | 0.4793 | 0.0246 | 0.0329 | 0.7559 | 45.00 | **0.0000** |
| MACD (Chong & Ng, 2008) | 0.7630 | 0.0380 | 0.0354 | 1.0753 | 85.00 | -0.0571 |
| GPT-3.5-Turbo (OpenAI, 2023) | 3.5539 | 0.1759 | 0.4913 | 0.3580 | 40.00 | -0.3810 |
| GPT-4.1-mini (Hurst et al., 2024) | 3.5733 | 0.1760 | 0.2626 | 0.6702 | 80.00 | -0.4661 |
| GPT-5-mini (OpenAI, 2025) | 7.3818 | 0.3573 | 0.3406 | 1.0490 | 90.00 | -0.0085 |
| LLaMa-3.1 (Dubey et al., 2024) | 2.0128 | 0.1016 | 0.6256 | 0.1623 | 75.00 | -1.9021 |
| Qwen-3 (Yang et al., 2025) | 5.4818 | 0.2679 | 0.3901 | 0.6867 | **100.00** | **0.0000** |
| DeepSeek-V3.2 (Liu et al., 2024) | 6.3699 | 0.3106 | 0.5320 | 0.5838 | 75.00 | -0.3546 |
| Gemini-3-Pro (Gemini Team, 2025) | 10.5248 | 0.5055 | 0.9061 | 0.5578 | 70.00 | -1.1459 |
| StockSim (Papadakis et al., 2025) *w/* GPT-5-mini | 4.2819 | 0.2109 | 0.4809 | 0.4386 | 90.00 | -0.6511 |
| ArenaTrader (Ma et al., 2025) *w/* GPT-5-mini | 6.1809 | 0.3008 | 0.3087 | 0.9745 | 80.00 | -0.2116 |
| ArenaTrader (Ma et al., 2025) *w/* GPT-4.1-mini | 4.0020 | 0.1968 | 0.2888 | 0.6813 | 75.00 | -0.2667 |
| DeMAgent *w/* GPT-3.5-Turbo | 5.3725 | 0.2627 | 0.3722 | 0.7056 | 80.00 | -0.2996 |
| DeMAgent *w/* GPT-4.1-mini | 6.1011 | 0.2976 | 0.4777 | 0.6231 | 80.00 | -0.4424 |
| DeMAgent *w/* GPT-5-mini | 9.3044 | 0.4475 | 0.5925 | 0.7553 | 85.00 | -0.1852 |
| DeMAgent *w/* LLaMa-3.1 | 6.9822 | 0.3395 | 0.5675 | 0.5983 | 80.00 | -0.3572 |
| DeMAgent *w/* Qwen-3 | 9.6524 | 0.4631 | 0.5305 | 0.8730 | 95.00 | -0.1402 |
| DeMAgent *w/* DeepSeek-V3.2 | 11.7127 | 0.5583 | **0.7935** | 0.7036 | 85.00 | -0.6579 |
| DeMAgent *w/* Gemini-3-Pro | **13.9459** | **0.6557** | 0.4110 | **1.5952** | 95.00 | **0.0000** |

MARKET, where prices are formed endogenously from agent interactions, preventing trajectory-level information leakage. Table 2 and Figure 4 summarize the performance of classical technical strategies, standalone LLMs, prior LLM-agent baselines, and DeMAgent. Standalone LLMs exhibit higher volatility and weaker risk-adjusted performance, while prior structured agents improve stability but deliver limited gains. In contrast, DeMAgent consistently achieves stronger performance across metrics. Figure 5 provides a representative trading visualization of DeMAgent *w/* GPT-5-mini on Stock A, illustrating price dynamics, executed actions, and portfolio wealth evolution. Notably, DeMAgent *w/* Gemini-3-Pro attains the highest TR (13.95%) and SR (1.60) with zero maximum drawdown, indicating effective risk control enabled by decoupling high-level planning from tool-based quantitative computation.

**Performance on Real Market Data.** We further evaluate DeMAgent on real-world equity (NASDAQ and CSI) and highly volatile CRYPTOCURRENCY markets. As shown in Table 3 and Figure 4, DeMAgent generally outperforms classical technical strategies, standalone LLMs, and prior LLM-agent baselines. Notably, it achieves balanced gains in profitability, risk-adjusted returns, and stability rather than optimizing a single metric. Its strong performance in CRYPTOCURRENCY markets further highlights robust

adaptation under extreme volatility.

### 4.3. Case & Ablation Studies and LLM Backbone Comparison

**LLM-Generated Quantitative Indicators in DECOUPLEDMARKET.** We present a representative case in DECOUPLEDMARKET (see Figure 6) illustrating emergent tool creation. The indicator is neither predefined by the designer nor explicitly solicited through prompts or action primitives; instead, it arises from closed-loop interaction between DeMAgent's reasoning process and endogenous market dynamics. In this example, the agent constructs an adaptive momentum-volatility indicator (AMVR) in response to evolving market conditions, rather than reusing a fixed template. Executed and evaluated entirely within DECOUPLEDMARKET, this case demonstrates that LLM-based agents can create and validate context-sensitive quantitative tools in a closed trading environment.

**Ablation Study.** Table 4 examines the effects of selectively disabling market analysis (MA), knowledge & tool creation (KTC), and evaluation & refinement (ER). Enabling MA or KTC alone improves performance over the standalone LLM baseline, with KTC contributing the largest gains, particularly in volatile CRYPTOCURRENCY markets. Combining MA and KTC yields further improvements, in-

*Table 3.* **Cross-market trading performance on real-world datasets.** Results on NASDAQ, CSI, and CRYPTOCURRENCY, comparing classical technical strategies, deep RL baselines, standalone LLMs, prior LLM-agent baselines, and DeMAgent. DeMAgent achieves the strongest overall performance and robustness across markets.

| Strategy | NASDAQ | | | CSI | | | CRYPTOCURRENCY | | |
|---|---|---|---|---|---|---|---|---|---|
| | TR ↑ | WR ↑ | SR ↑ | TR ↑ | WR ↑ | SR ↑ | TR ↑ | WR ↑ | SR ↑ |
| Buy & Hold | -0.38 | 48.80 | -0.001 | 26.52 | 55.90 | 0.26 | 13.76 | 60.00 | 0.092 |
| ATR (Wilder, 1978) | 0.60 | 22.20 | 0.02 | 4.54 | 25.40 | 0.05 | -4.79 | 30.90 | -0.02 |
| RSI (Wilder, 1978) | -5.06 | 4.90 | -0.21 | 2.30 | 3.40 | 0.10 | 7.30 | 10.90 | 0.21 |
| ZMR (Poterba & Summers, 1988) | 0.13 | 12.20 | 0.01 | 3.75 | 6.80 | 0.11 | 13.57 | 18.20 | 0.11 |
| Bollinger Bands (Bollinger, 1992) | -0.06 | 0.00 | -0.16 | 0.00 | 0.00 | 0.00 | 8.54 | 3.60 | 0.00 |
| SMA (Gencay, 1996) | -6.47 | 17.10 | -0.22 | 21.11 | 35.60 | 0.18 | -4.79 | 23.60 | -0.01 |
| MACD (Chong & Ng, 2008) | -5.71 | 29.30 | -0.13 | 27.24 | 49.20 | 0.21 | 2.99 | 40.00 | 0.03 |
| DDPG (Lillicrap et al., 2015) | 1.94 | 51.22 | 0.05 | 12.74 | 55.93 | 0.25 | 3.76 | 58.18 | 0.04 |
| PPO (Schulman et al., 2017) | -5.18 | 46.34 | -0.05 | 16.13 | 50.85 | 0.18 | 2.93 | 56.36 | 0.03 |
| A2C (Haarnoja et al., 2018) | -5.47 | 46.34 | -0.04 | 22.92 | 55.93 | 0.22 | 4.78 | 56.36 | 0.05 |
| TD3 (Fujimoto et al., 2018) | 1.93 | 48.78 | 0.04 | 32.60 | 50.85 | 0.20 | 4.78 | 56.36 | 0.05 |
| GPT-3.5-Turbo (OpenAI, 2023) | -2.13 | 35.71 | -0.16 | 4.31 | 45.00 | 0.08 | 2.78 | 35.71 | 0.03 |
| GPT-4.1-mini (Hurst et al., 2024) | 3.27 | 54.76 | 0.11 | 18.36 | 25.00 | 0.28 | 0.73 | 50.00 | 0.02 |
| GPT-5-mini (OpenAI, 2025) | 1.20 | 57.14 | 0.03 | 20.05 | 53.33 | 0.25 | 4.21 | 58.93 | 0.07 |
| LLaMa-3.1 (Dubey et al., 2024) | -4.89 | 42.86 | -0.15 | 8.41 | 48.33 | 0.13 | -2.69 | 57.14 | -0.01 |
| Qwen-3 (Yang et al., 2025) | 3.36 | 57.14 | 0.09 | 12.94 | 58.33 | 0.22 | 2.18 | 53.57 | 0.11 |
| DeepSeek-V3.2 (Liu et al., 2024) | -2.41 | 52.38 | -0.03 | 21.44 | 58.33 | 0.26 | 14.75 | 60.71 | 0.13 |
| Gemini-3-Pro (Gemini Team, 2025) | -0.66 | **59.52** | -0.05 | 12.48 | 55.00 | 0.15 | 4.67 | 55.36 | 0.05 |
| StockSim (Papadakis et al., 2025) *w/* GPT-5-mini | -0.16 | 53.97 | -0.02 | 10.33 | 55.36 | 0.13 | 11.20 | 40.00 | 0.12 |
| ArenaTrader (Ma et al., 2025) *w/* GPT-4.1-mini | 1.04 | 54.76 | 0.04 | 6.70 | **58.33** | **0.31** | 9.22 | 58.93 | 0.08 |
| ArenaTrader (Ma et al., 2025) *w/* GPT-5-mini | -7.57 | 40.48 | -0.19 | 7.11 | 51.67 | 0.25 | 13.85 | 62.50 | 0.15 |
| DeMAgent *w/* GPT-5-mini | **8.49** | **59.52** | **0.21** | **41.32** | 55.00 | 0.29 | **25.21** | **67.86** | **0.29** |

*Table 4.* **Module ablation of DeMAgent across markets.** Evaluation of market analysis (MA), knowledge & tool creation (KTC), and evaluation & refinement (ER) by selectively enabling each module. The full model achieves the most stable and competitive performance across markets.

| Model | Ablation | | | NASDAQ | | | CSI | | | CRYPTOCURRENCY | | |
|---|---|---|---|---|---|---|---|---|---|---|---|---|
| | MA | KTC | ER | TR ↑ | WR ↑ | SR ↑ | TR ↑ | WR ↑ | SR ↑ | TR ↑ | WR ↑ | SR ↑ |
| GPT-5-mini (OpenAI, 2025) | ○ | ○ | ○ | 1.20 | 57.14 | 0.03 | 16.78 | 60.00 | 0.25 | 5.19 | 58.93 | 0.07 |
| DeMAgent (MA) | ● | ○ | ○ | 1.63 | 52.38 | 0.03 | 17.17 | **65.00** | 0.21 | 9.36 | 53.57 | 0.13 |
| DeMAgent (KTC) | ○ | ● | ○ | 3.19 | 47.62 | 0.10 | 19.46 | 60.00 | 0.22 | 20.96 | 61.11 | 0.20 |
| DeMAgent (MA + KTC) | ● | ● | ○ | 5.14 | 45.24 | 0.11 | 29.64 | 58.33 | 0.28 | 17.30 | 62.50 | 0.16 |
| DeMAgent (Full) | ● | ● | ● | **8.49** | **59.52** | **0.21** | **41.32** | 55.00 | **0.29** | **25.21** | **67.86** | **0.29** |

dicating that structured market reasoning and tool-based quantitative execution are complementary. With all modules enabled, DeMAgent achieves the strongest and most stable performance across markets, and stronger backbones generally yield better returns and risk-adjusted performance.

**Effect of LLM Backbone Across Real Markets.** We evaluate DeMAgent with seven LLM backbones on three real-world markets. Table 5 reports TR (total return) and SR (Sharpe ratio) for each backbone. GPT-5-mini achieves the highest TR and SR in most markets, indicating that the DeMAgent pipeline can effectively leverage stronger language models to improve profitability and risk-adjusted performance. Other backbones exhibit more variable results, highlighting the sensitivity of trading outcomes to the underlying LLM across market regimes.

*Table 5.* **Effect of LLM backbones on DeMAgent performance.** Results across NASDAQ, CSI, and CRYPTOCURRENCY show that stronger backbones generally yield higher returns and improved risk-adjusted performance.

| Backbone | NASDAQ | | CSI | | CRYPTO | |
|---|---|---|---|---|---|---|
| | TR ↑ | SR ↑ | TR ↑ | SR ↑ | TR ↑ | SR ↑ |
| LLaMa-3.1 | -1.59 | -0.04 | 13.17 | 0.25 | 6.24 | 0.05 |
| Qwen-3 | 4.96 | 0.11 | 35.62 | 0.22 | 13.34 | 0.14 |
| DeepSeek-V3.2 | 4.15 | 0.09 | 15.41 | 0.17 | **26.93** | 0.19 |
| Gemini-3-Pro | 3.75 | 0.13 | 29.36 | 0.26 | 16.27 | 0.11 |
| GPT-3.5-Turbo | -0.71 | -0.02 | 16.31 | 0.27 | 14.83 | 0.11 |
| GPT-4.1-mini | 2.97 | 0.10 | 16.69 | 0.18 | 16.49 | 0.12 |
| GPT-5-mini | **8.49** | **0.21** | **41.32** | **0.29** | 25.21 | **0.29** |

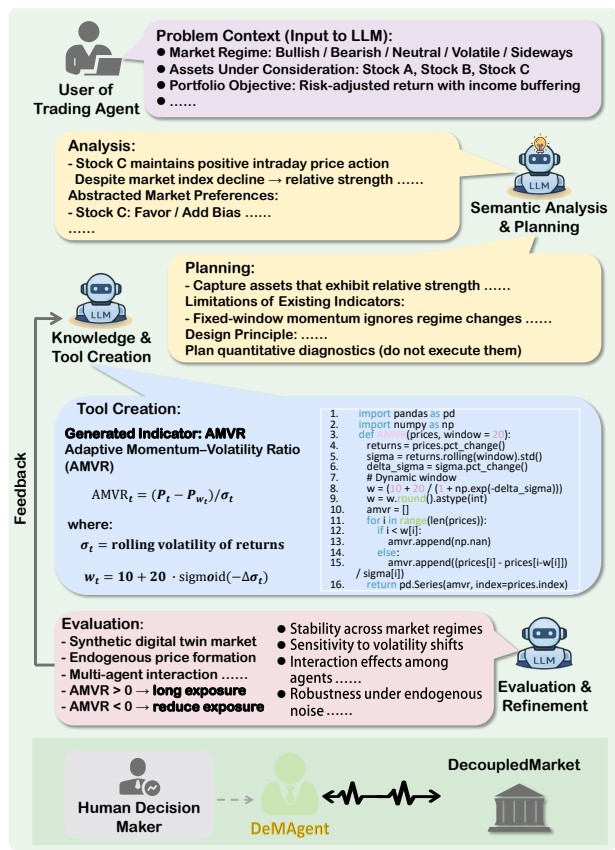

*Figure 6.* **Market-driven indicator generation and validation in DECOUPLEDMARKET.** DeMAgent converts market analysis into an adaptive momentum-volatility indicator (AMVR), generates executable code, and validates the resulting strategy within the digital twin market.

## 5. Conclusion

We propose a tool-creation paradigm for LLM-driven quantitative reasoning, where models dynamically synthesize and organize task-specific analytical tools within a masked reasoning framework. To enable controlled, reproducible evaluation of the resulting tools, we introduce DECOUPLED-MARKET, a decoupled digital twin market. Building on this environment, we develop DeMAgent, an LLM-driven analysis agent that performs structured planning, tool orchestration, and result interpretation without directly executing trades. Experiments show that reasoning behaviors developed through self-play in DECOUPLEDMARKET exhibit bounded practical adaptability: the method learns reasoning strategies that remain useful across several materially different real-world market settings, without implying validity for all markets or institutional contexts. Controlled experiments establish the causal role of tool creation, evaluation, and refinement (mechanism validation), while real-world data testing validates practical utility under realistic, non-stationary conditions. Overall, our framework offers a flexible and

testable approach for integrating high-level LLM reasoning with the creation and evaluation of quantitative tools.

## Acknowledgements

This work was supported in part by the National Natural Science Foundation of China under Grants 62301213 and 62271361, the Hubei Provincial Key Research and Development Program under Grant 2024BAB039, the Open Project of the Hubei Key Laboratory of Big Data Intelligent Analysis and Application, Hubei University under Grant 2024BDIAA01, the Shanghai Science and Technology Program under Grant 25HB2703100, and the National Research Foundation, Singapore and Infocomm Media Development Authority under its Trust Tech Funding Initiative. Any opinions, findings, conclusions, or recommendations expressed in this material are those of the author(s) and do not reflect the views of the National Research Foundation, Singapore or Infocomm Media Development Authority. The authors acknowledge Beijing PARATERA Technology Co., LTD for providing high-performance AI computing resources.

## Impact Statement

This work studies a framework for LLM-based quantitative reasoning and evaluates it through historical backtesting on real-world financial datasets and controlled experiments in a digital twin market. It aims to improve the reliability and interpretability of LLM-based decision-making by decoupling high-level semantic reasoning from numerical computation and exposing intermediate reasoning artifacts for inspection. All experiments use historical data and simulated market interactions, with no deployment in live trading systems or interaction with real financial markets. Accordingly, the proposed system is intended solely for research on quantitative reasoning, rather than real-time automated trading or market intervention.

Financial models can exhibit performative effects, where deployment changes the market dynamics the model was designed to exploit (MacKenzie, 2008). As increasingly capable LLM-based trading systems emerge, large-scale adoption could introduce systemic feedback loops, amplify correlated behaviors, and propagate unanticipated failure modes. We therefore emphasize that DeMAgent is a research framework for studying LLM-based quantitative reasoning under controlled conditions, rather than a production-ready trading system. To discourage direct real-world deployment, we intentionally abstract away infrastructure required for live execution and market connectivity. More broadly, the societal impacts of this work align with those of machine learning systems for decision-making under uncertainty, including the benefits of improved reasoning transparency and the risks of misuse or over-reliance.

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

## A. Self-Play: Formal Definition

We define **self-play** in our context as a *synthetic form of play*: a single-agent, open-ended interaction process under realistic uncertainty, rather than the classical multi-agent reinforcement learning setting. Specifically, an LLM-based agent iteratively refines its decision-making and tool usage through repeated interactions with a dynamic, non-stationary market environment, where other participants follow fixed or heuristic strategies and thereby create a largely unpredictable environment.

Unlike classical self-play in multi-agent RL, *e.g.*, AlphaGo, DeMAgent's self-play is single-agent and performance-driven, progressively refining tools and strategies based on metrics such as total return and Sharpe ratio. Although optimization is guided by performance metrics, overfitting is mitigated by evaluation across multiple markets and time periods.

## B. Additional Details of the Market Protocol

### B.1. Order Matching and Execution

At each timestep, agents submit portfolio-level buy or sell quantities for each asset. Orders are aggregated per asset and executed by a centralized matching engine. To abstract away order-book microstructure while preserving aggregate supply–demand effects, DECOUPLEDMARKET uses a volume-conserving execution rule: when aggregate buy and sell volumes are imbalanced, executed quantities are scaled proportionally so that total executed buy volume equals total executed sell volume. All trades clear at a common reference price $\tilde{P}^t_{\text{close},n}$, which serves as the notional clearing price at timestep $t$. Excess demand $\Delta^t_n$ is computed from executed, rather than submitted, quantities, ensuring that price impact reflects realized trades rather than unfulfilled intentions.

### B.2. Non-Stationarity Induced by Agent Adaptation

Although the environment transition function $\mathcal{T}$ is stationary, effective non-stationarity arises from adaptive agent behavior. In our experiments, DeMAgent updates its reasoning traces, tool selection, and decision abstractions across episodes via self-feedback, while baseline agents are either fixed or periodically re-optimized depending on the setting. From the perspective of any individual agent, the joint policy of other agents $\pi_{-i}$ therefore evolves, inducing a non-stationary decision process despite fixed market mechanics.

### B.3. Deterministic Trade Execution and Endogenous Price Update

In DECOUPLEDMARKET, agents do not submit explicit orders. Instead, each agent provides a portfolio-level allocation action $a^i_t \in \mathbb{R}^N$, which is deterministically mapped to executed buy and sell quantities by a centralized matching engine:

$$\left\{ q^{\text{buy},t}_{n,i}, q^{\text{sell},t}_{n,i} \right\}^N_{n=1} = \mathcal{M}_n \left( a^i_t, s_t \right). \tag{9}$$

The matching mechanism $\mathcal{M}_n(\cdot)$ is fixed, agent-agnostic, and contains no adaptive or learned components. Its sole role is to translate portfolio-level decisions into realized trades for computing aggregate order-flow statistics.

Executed trades induce excess demand $\Delta^t_n$, which summarizes the net order-flow imbalance within each timestep. Price impact is computed as a bounded, monotone function of excess demand scaled by recent trading activity:

$$\text{Impact}^t_n = \kappa \cdot \tanh \left( \frac{\Delta^t_n}{L^t_n} \right), \quad L^t_n = \sum_{\tau=t-T}^{t} V^\tau_n, \tag{10}$$

where $V^\tau_n$ denotes realized trading volume. The resulting transaction price is:

$$P^t_{\text{trade},n} = \tilde{P}^t_{\text{close},n} \cdot \left( 1 + \text{Impact}^t_n \right). \tag{11}$$

Rather than modeling full market microstructure, the reference price for the next timestep is obtained by aggregating contemporaneous transactional statistics:

$$P^{t+1}_n = f \left( P^t_{\text{trade},n}, P^t_{\text{vwap},n}, P^t_{\text{trend},n} \right), \tag{12}$$

*Table 6.* **LLM backbones evaluated in this work.** Publicly available characteristics are reported where applicable.

| Model Identifier | Year | High-Level Description | Data Cutoff |
|---|---|---|---|
| GPT-3.5-Turbo | 2023 | Chat-optimized proprietary LLM | Sep. 2021 |
| GPT-4.1-mini | 2024 | Lightweight multimodal GPT-4-class model | Unknown |
| GPT-5-mini | 2025 | API-accessible next-generation GPT-series variant | Unknown |
| Meta-LLaMa-3.1-70B-Instruct-Turbo | 2024 | Open-weight instruction-tuned LLM (70B) | Dec. 2023 |
| Qwen3-32B | 2025 | Open-weight multilingual LLM (32B) | Unknown |
| DeepSeek-V3.2-Exp | 2025 | Experimental MoE-based LLM, API-accessible | Unknown |
| Gemini-3-Pro-Preview | 2025 | Proprietary LLM, preview version | Unknown |

where

$$P_{\text{vwap},n}^t = \frac{\sum_{\tau=t-T}^{t} P_{\text{trade},n}^\tau V_n^\tau}{\sum_{\tau=t-T}^{t} V_n^\tau}, \tag{13}$$

$$P_{\text{trend},n}^t = P_{\text{trade},n}^t - P_{\text{vwap},n}^t, \tag{14}$$

where $P_{\text{vwap},n}^t$ captures volume-weighted transactional consensus and $P_{\text{trend},n}^t$ measures short-horizon price pressure induced by order flow.

The aggregation function $f(\cdot)$ is deterministic, fixed across assets, and independent of agent identities, rewards, or learning signals. It encodes no directional price prediction, trend continuation, or mean-reversion assumptions; thus, any exploitable structure arises endogenously from agent interactions and order-flow dynamics rather than being imposed by the price update rule.

## C. DECOUPLEDMARKET Agent Configuration and Large Language Model Details

### C.1. DECOUPLEDMARKET Agent Configuration

DECOUPLEDMARKET hosts a heterogeneous population of trading agents operating within a unified simulation environment. The agent population comprises three broad categories: (1) classical technical strategy agents that implement rule-based heuristics, including Buy-and-Hold, MACD, RSI, SMA, and Bollinger Bands; (2) standalone LLM agents instantiated with different LLM backbones, such as Qwen3-32B, DeepSeek-V3.2, LLaMa-3.1-70B, Gemini-3-Pro, GPT-4.1-mini, and GPT-5-mini; and (3) LLM-agent baselines, including existing implementations, *i.e.*, ArenaTrader and StockSim, as well as our proposed LLM-driven agent, DeMAgent, which augments a base LLM with explicit tool-creation and invocation capabilities.

All agents interact with the same market simulator and share an identical observation space, action space, and execution protocol. In particular, LLM-based agents differ only in their underlying language model backbones, while the surrounding agent interface, prompt structure, and tool access are held fixed across models. No model-specific fine-tuning or task-specific adaptation is applied. This design ensures that observed behavioral differences can be attributed to the LLM backbone itself rather than to variations in system configuration.

### C.2. Evaluated Large Language Models

This section summarizes the LLMs used as backbones for LLM-based agents in our experiments. The selected models are intended to reflect the diversity of contemporary LLM ecosystems rather than to optimize performance for any specific trading strategy or market condition. Accordingly, the model set includes both proprietary and open-weight models accessed via standard inference APIs, covering chat-oriented models, multimodal variants, lightweight or efficiency-focused versions, and preview or experimental releases.

Table 6 lists all evaluated LLM backbones, together with their platform-provided identifiers, approximate release years, and high-level descriptions. For each model, we report the training data cutoff when officially disclosed by the corresponding provider. When such information is not publicly available, it is conservatively marked as unknown. Model identifiers are reported verbatim as provided by the respective platforms at the time of experimentation. We make no assumptions about undisclosed training procedures, proprietary data sources, or internal architectural details beyond what is publicly documented. This reporting strategy supports transparency and reproducibility at the level of model access while respecting

*Table 7.* **Testing periods for each market benchmark.** Evaluation windows used for CRYPTOCURRENCY, NASDAQ, and CSI.

| Market | Start Date | End Date |
|---|---|---|
| CRYPTOCURRENCY | Sep. 1, 2025 | Oct. 26, 2025 |
| NASDAQ | Oct. 1, 2025 | Nov. 30, 2025 |
| CSI | Aug. 1, 2025 | Oct. 31, 2025 |

the disclosure boundaries of different providers.

## D. Dataset Details

### D.1. Dataset Overview

Experiments are conducted on three market benchmarks: CRYPTOCURRENCY, NASDAQ, and CSI, with evaluation periods summarized in Table 7. For each benchmark, the set of tradable assets remains fixed throughout the evaluation horizon to ensure consistent comparisons across agents and over time. The CRYPTOCURRENCY benchmark includes Bitcoin (BTC), Solana (SOL), and Binance Coin (BNB). The NASDAQ benchmark comprises Apple Inc. (AAPL), Microsoft Corporation (MSFT), and NVIDIA Corporation (NVDA). The CSI benchmark consists of Kweichow Moutai (sh.600519), Contemporary Amperex Technology Co., Limited (sz.300750), and LONGi Green Energy Technology Co., Ltd. (sh.601012).

Market data are obtained from YAHOO FINANCE for the CRYPTOCURRENCY and NASDAQ benchmarks, and from BAOSTOCK for the CSI benchmark. For each asset, the dataset includes daily open, high, low, and close prices, trading volume, and commonly used technical indicators derived from historical prices. All benchmarks follow a unified data schema and are aligned with the DECOUPLEDMARKET environment.

### D.2. Data Processing

All simulations exclude weekends and official market holidays. Missing values are handled using forward filling based solely on past observations, thereby avoiding look-ahead bias. Technical indicators, including moving averages, momentum measures, and volatility statistics, are computed from daily price series and provided to agents as part of the observable market state. Overall, the processed datasets provide a consistent, leakage-free representation of market dynamics suitable for evaluating multi-agent portfolio management strategies.

## E. Implementation Details and Hyperparameters

All LLM-based agents are evaluated under identical environmental settings, market data, and testing periods. Except for the underlying LLM backbone, all components of the agent architecture are held fixed to ensure fair and controlled comparisons.

We compare DeMAgent against a diverse set of baselines spanning classical technical trading rules, reinforcement learning methods, and simulation-based multi-agent systems:

**(1) Buy & Hold**: A passive baseline that maintains fixed asset positions throughout the evaluation period.

**(2) SMA** (Gencay, 1996): A simple moving average crossover strategy based on short- and long-term price trends.

**(3) ZMR** (Poterba & Summers, 1988): A zone-based mean-reversion strategy that trades on price deviations from predefined bounds.

**(4) MACD** (Chong & Ng, 2008): A momentum-based strategy using standard MACD and signal line crossovers.

**(5) PPO** (Schulman et al., 2017): Proximal Policy Optimization (Schulman et al., 2017), an on-policy actor-critic reinforcement learning method.

**(6) A2C** (Haarnoja et al., 2018): Advantage Actor-Critic (Haarnoja et al., 2018), an on-policy actor-critic method that jointly learns policy and value functions.

**(7) TD3** (Fujimoto et al., 2018): Twin Delayed Deep Deterministic Policy Gradient (Fujimoto et al., 2018), an off-policy actor-critic method with delayed policy updates.

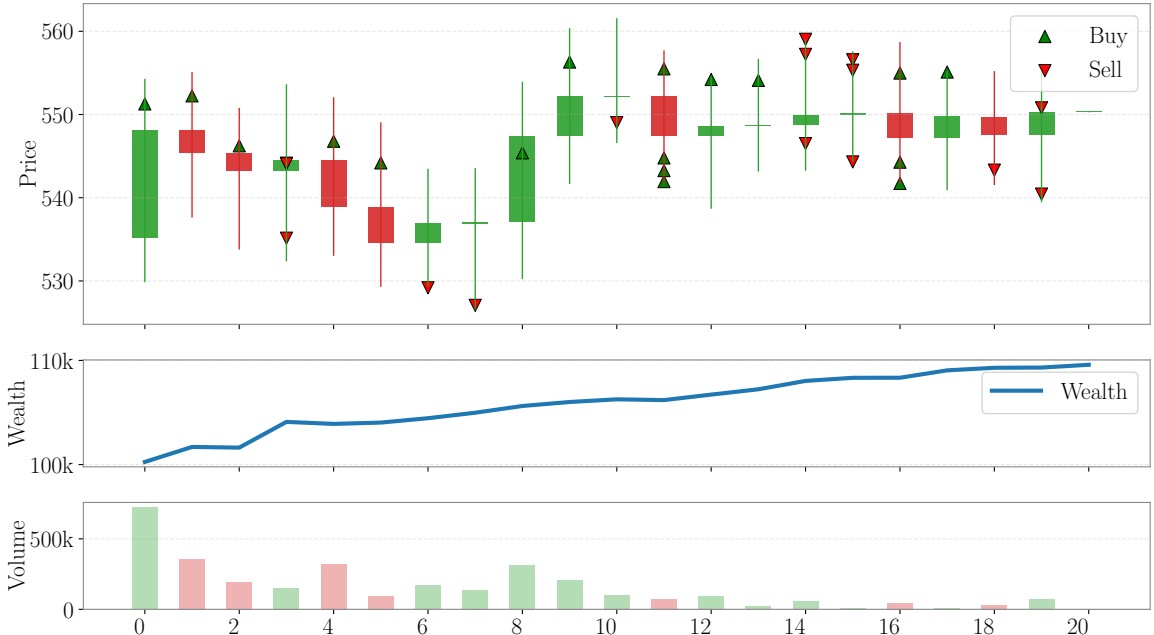

*Figure 7.* **Trading dynamics and portfolio evolution for Stock B in DECOUPLEDMARKET.** Candlestick prices are overlaid with executed buy and sell actions, portfolio wealth trajectories, and trading volume, illustrating the interaction between agent decisions and endogenous market dynamics.

**(8) DDPG** (Lillicrap et al., 2015): Deep Deterministic Policy Gradient (Lillicrap et al., 2015), an off-policy actor-critic method for continuous control.

**(9) StockSim** (Papadakis et al., 2025): A simulation-based multi-agent trading system that models interactions among heterogeneous agents.

**(10) ArenaTrader** (Ma et al., 2025): A multi-agent trading framework serving as a system-level baseline for comparison.

## F. Detailed Evaluation of DECOUPLEDMARKET and Real-World Market

### F.1. Visualization Analysis in DECOUPLEDMARKET

Figures 7 and 8 illustrate trading behaviors for Stock B and Stock C in DECOUPLEDMARKET, visualizing executed trades, portfolio wealth evolution, and market volume.

**Stock B Analysis.** As shown in Figure 7, Stock B exhibits moderate short-term fluctuations with an overall upward trend. DeMAgent responds adaptively by executing buy actions near local drawdowns and sell actions near local peaks. Portfolio wealth increases steadily with limited drawdowns, indicating effective capture of short-term price movements. Trading volume is higher in the early stages, reflecting active position establishment, and gradually stabilizes as positions are consolidated.

**Stock C Analysis.** As shown in Figure 8, Stock C displays higher volatility with frequent oscillations around a relatively flat trend. In response, DeMAgent adopts a more conservative strategy, executing fewer trades and making smaller position adjustments. Wealth accumulation is smoother and slower than in Stock B, reflecting cautious exposure under elevated uncertainty. Trading volume remains lower and more uniform, indicating controlled engagement with volatile conditions.

Overall, these visualizations demonstrate that DeMAgent dynamically adjusts trading frequency and position sizing in response to market characteristics, balancing profit opportunities against risk exposure. Quantitative comparisons across NASDAQ, CSI, and CRYPTOCURRENCY markets are reported in Tables 9 and 10 and **??**, covering reinforcement learning baselines, standalone LLM agents, existing LLM-based trading systems, and DeMAgent instantiated with different LLM

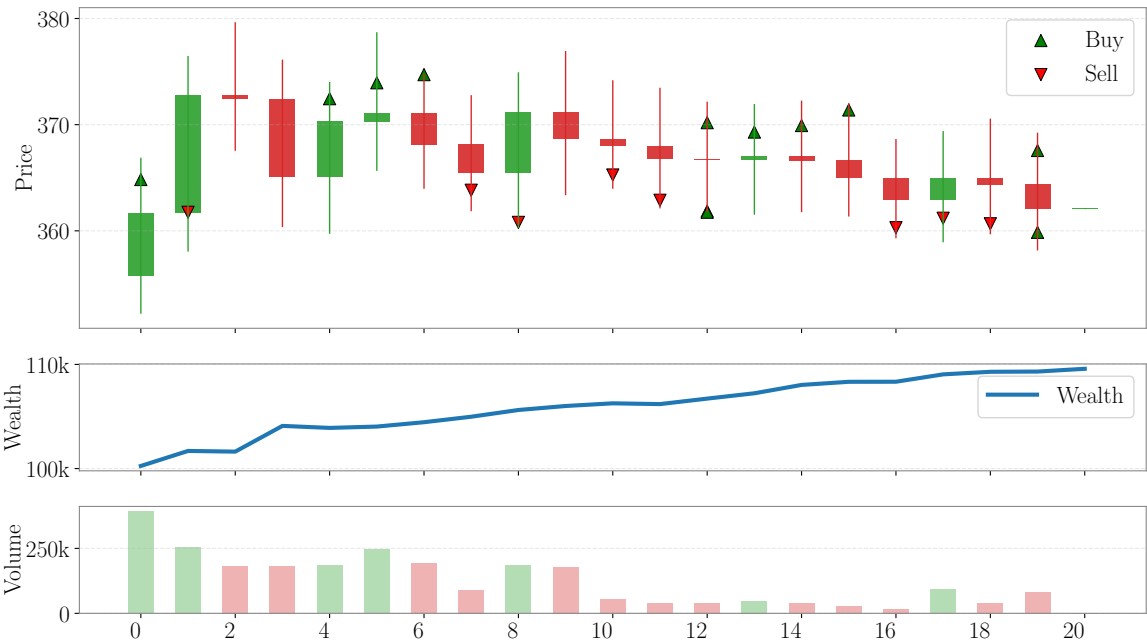

*Figure 8.* **Trading dynamics and portfolio evolution for Stock C in DECOUPLEDMARKET.** Candlestick prices are overlaid with executed buy and sell actions, portfolio wealth trajectories, and trading volume, highlighting behavioral consistency across different market conditions.

*Table 8.* **Computational cost and latency of different LLM agents.** All results are measured on the CSI market from Sep. 3, 2025 to Oct. 31, 2025. Success rate denotes the proportion of trading decisions that pass all hard-constraint validations.

| Agent | Calls | Success (%) | Latency (s) | Total ($) | Per-Call ($) |
|---|---|---|---|---|---|
| GPT-5-mini | 36 | 100.0 | 5.2 | 0.022 | 0.0006 |
| GPT-4.1-mini | 36 | 100.0 | 4.8 | 0.018 | 0.0005 |
| StockSim *w/* GPT-5-mini | 72 | 100.0 | 8.9 | 0.044 | 0.0006 |
| ArenaTrader *w/* GPT-5-mini | 108 | 99.6 | 12.3 | 0.066 | 0.0006 |
| DeMAgent *w/* GPT-5-mini | 164 | 99.6 | 19.8 | 0.098 | 0.0006 |

backbones.

### F.2. Computational Cost and Efficiency Analysis

To assess the computational overhead of dynamic tool creation, we conduct detailed efficiency profiling on the CSI market from Sep. 3, 2025 to Oct. 31, 2025. Table 8 reports LLM API calls, success rates, latency, and cost for DeMAgent and baseline agents.

### F.3. Quantitative Evaluation of Real-World Market Performance

Tables 9 and 10 and **??** summarize performance on NASDAQ, CSI, and CRYPTOCURRENCY. Reported metrics include total return (TR), portfolio return $R_p$, portfolio volatility $\sigma_p$, Sharpe ratio (SR), win rate (WR), maximum drawdown (MD), and Calmar ratio (CR). Consistent with the main-text summary, DeMAgent achieves strong and stable performance across all markets. On NASDAQ and CSI, it generally outperforms deep reinforcement learning baselines and standalone LLMs in both absolute and risk-adjusted metrics, indicating effective adaptation to moderately trending environments. Improvements in Sharpe and Calmar ratios further suggest that the gains are not driven solely by increased risk-taking.

On CRYPTOCURRENCY, DeMAgent maintains high risk-adjusted returns while controlling maximum drawdown, demonstrating robustness under extreme volatility and pronounced non-stationarity. Compared with alternative approaches, the agent exhibits stronger adaptive reasoning and improved awareness of multi-agent interactions in rapidly changing markets.

*Table 9.* **Market-specific performance comparison on NASDAQ.** Comparison of deep RL baselines, standalone LLMs, LLM-based trading agents, and `DeMAgent` with different LLM backbones. Metrics include TR, $R_p$, $\sigma_p$, SR, WR, MD, and CR.

| Model | TR ↑ | $R_p$ ↑ | $\sigma_p$ ↓ | SR ↑ | WR ↑ | MD ↑ | CR ↑ |
|---|---|---|---|---|---|---|---|
| DDPG (Lillicrap et al., 2015) | 1.9422 | 0.0518 | 0.9961 | 0.0520 | 51.22 | -6.3966 | 1.9620 |
| PPO (Schulman et al., 2017) | -5.1777 | -0.1077 | 2.1171 | -0.0509 | 46.34 | -13.4931 | -2.0659 |
| A2C (Haarnoja et al., 2018) | -5.4689 | -0.1056 | 2.5385 | -0.0416 | 46.34 | -14.5093 | -2.0143 |
| TD3 (Fujimoto et al., 2018) | 1.9343 | 0.0583 | 1.5408 | 0.0379 | 48.78 | -7.0234 | 1.7793 |
| GPT-3.5-Turbo (OpenAI, 2023) | -2.1265 | -0.0507 | 0.3085 | -0.1643 | 35.71 | -2.3424 | -5.1654 |
| GPT-4.1-mini (Hurst et al., 2024) | 3.2666 | 0.0790 | 0.7105 | 0.1112 | 54.76 | -3.2392 | 6.5671 |
| GPT-5-mini (OpenAI, 2025) | 1.1951 | 0.0328 | 0.9605 | 0.0342 | 57.14 | -5.7651 | 1.2815 |
| LLaMa-3.1 (Dubey et al., 2024) | -4.8882 | -0.1162 | 0.7847 | -0.1481 | 42.86 | -6.3248 | -4.1061 |
| Qwen-3 (Yang et al., 2025) | 3.3604 | 0.0831 | 0.9463 | 0.0878 | 57.14 | -4.4336 | 4.9472 |
| DeepSeek-V3.2 (Liu et al., 2024) | -2.4070 | -0.0446 | 1.6572 | -0.0269 | 52.38 | -9.4450 | -1.4399 |
| Gemini-3-Pro (Gemini Team, 2025) | -0.6609 | -0.0154 | **0.2820** | -0.0546 | **59.52** | -1.8522 | -2.1058 |
| StockSim (Papadakis et al., 2025) *w/* GPT-5-mini | -0.1600 | -0.3100 | 0.0705 | -0.0174 | 53.97 | -1.1700 | -0.2684 |
| ArenaTrader (Ma et al., 2025) *w/* GPT-4.1-mini | 1.0384 | 0.0267 | 0.6603 | 0.0405 | 54.76 | -4.1042 | 1.5580 |
| ArenaTrader (Ma et al., 2025) *w/* GPT-5-mini | -7.5708 | -0.1829 | 0.9470 | -0.1931 | 40.48 | -9.5480 | -3.9429 |
| DeMAgent *w/* LLaMa-3.1 | -1.5934 | -0.0340 | 0.9342 | -0.0364 | 47.62 | -4.7435 | -1.9369 |
| DeMAgent *w/* Qwen-3 | 4.9614 | 0.1211 | 1.0859 | 0.1115 | **59.52** | -4.3911 | 7.6779 |
| DeMAgent *w/* Gemini-3-Pro | 3.7546 | 0.0902 | 0.7027 | 0.1284 | 54.76 | -3.0784 | 8.0401 |
| DeMAgent *w/* DeepSeek-V3.2 | 4.1506 | 0.1036 | 1.1693 | 0.0886 | 57.14 | -5.4750 | 5.0476 |
| DeMAgent *w/* GPT-3.5-Turbo | -0.7064 | -0.0135 | 0.8341 | -0.0162 | 45.24 | -6.1078 | -0.6818 |
| DeMAgent *w/* GPT-4.1-mini | 2.9657 | 0.0723 | 0.7395 | 0.0977 | 54.76 | **-2.4525** | 7.8150 |
| DeMAgent *w/* GPT-5-mini | **8.4876** | **0.1986** | 0.9586 | **0.2072** | **59.52** | -3.0585 | **20.6093** |

*Table 10.* **Market-specific performance comparison on CSI.** Comparison of deep RL baselines, standalone LLMs, LLM-based trading agents, and `DeMAgent` with different LLM backbones. Metrics include TR, $R_p$, $\sigma_p$, SR, WR, MD, and CR.

| Model | TR ↑ | $R_p$ ↑ | $\sigma_p$ ↓ | SR ↑ | WR ↑ | MD ↑ | CR ↑ |
|---|---|---|---|---|---|---|---|
| DDPG (Lillicrap et al., 2015) | 12.7433 | 0.2067 | 0.8133 | 0.2542 | 55.93 | -2.5110 | 26.6481 |
| PPO (Schulman et al., 2017) | 16.1287 | 0.2638 | 1.4333 | 0.1840 | 50.85 | -6.5126 | 13.7264 |
| A2C (Haarnoja et al., 2018) | 22.9204 | 0.3639 | 1.6673 | 0.2183 | 55.93 | -5.9422 | 23.8019 |
| TD3 (Fujimoto et al., 2018) | 32.6005 | 0.5102 | 2.5344 | 0.2013 | 50.85 | -10.0000 | 23.3744 |
| GPT-3.5-Turbo (OpenAI, 2023) | 4.3079 | 0.0750 | 0.9770 | 0.0768 | 45.00 | -5.5948 | 3.4640 |
| GPT-4.1-mini (Hurst et al., 2024) | 18.3619 | 0.2891 | 1.2711 | 0.2275 | 25.00 | -4.9317 | 20.8848 |
| GPT-5-mini (OpenAI, 2025) | 20.0472 | 0.3125 | 1.2425 | 0.2515 | 53.33 | -4.0100 | 28.7822 |
| LLaMa-3.1 (Dubey et al., 2024) | 8.4076 | 0.1404 | 1.0827 | 0.1297 | 48.33 | -4.3224 | 9.3380 |
| Qwen-3 (Yang et al., 2025) | 12.9411 | 0.1912 | 0.8532 | 0.2241 | **58.33** | **-0.8039** | 31.8079 |
| DeepSeek-V3.2 (Liu et al., 2024) | 21.4357 | 0.3323 | 1.2843 | 0.2587 | **58.33** | -5.4629 | 23.0788 |
| Gemini-3-Pro (Gemini Team, 2025) | 12.4821 | 0.2055 | 1.3763 | 0.1493 | 55.00 | -4.9120 | 13.0067 |
| StockSim (Papadakis et al., 2025) *w/* GPT-5-mini | 10.3300 | 0.1850 | 1.3852 | 0.1336 | 55.36 | -6.0908 | 9.1351 |
| ArenaTrader (Ma et al., 2025) *w/* GPT-4.1-mini | 6.7009 | 0.1088 | **0.3546** | 0.3067 | **58.33** | -0.9737 | 32.1584 |
| ArenaTrader (Ma et al., 2025) *w/* GPT-5-mini | 7.1085 | 0.1155 | 0.4536 | 0.2547 | 51.67 | -0.9367 | 35.6906 |
| DeMAgent *w/* LLaMa-3.1 | 13.1668 | 0.2098 | 0.8401 | 0.2497 | 55.00 | -2.8082 | 24.2580 |
| DeMAgent *w/* Qwen-3 | 35.6180 | 0.5383 | 2.4694 | 0.2180 | 53.33 | -9.9657 | 26.0421 |
| DeMAgent *w/* Gemini-3-Pro | 29.3618 | 0.4445 | 1.7291 | 0.2571 | **58.33** | -6.3736 | 30.5696 |
| DeMAgent *w/* DeepSeek-V3.2 | 15.4085 | 0.2502 | 1.5025 | 0.1666 | **58.33** | -8.4165 | 9.8089 |
| DeMAgent *w/* GPT-3.5-Turbo | 16.3123 | 0.2566 | 0.9487 | 0.2704 | 53.33 | -2.9887 | 29.6578 |
| DeMAgent *w/* GPT-4.1-mini | 16.6878 | 0.2688 | 1.5218 | 0.1767 | **58.33** | -6.6718 | 13.6707 |
| DeMAgent *w/* GPT-5-mini | **41.3224** | **0.5982** | 2.0484 | **0.2920** | 55.00 | -4.3074 | **76.0203** |

Overall, these results indicate that the advantages of `DeMAgent` are consistent across heterogeneous markets and are not specific to any single asset class or LLM backbone.

*Table 11.* **Market-specific performance comparison on CRYPTOCURRENCY.** Comparison of deep RL baselines, standalone LLMs, LLM-based trading agents, and DeMAgent with different LLM backbones. Metrics include TR, $R_p$, $\sigma_p$, SR, WR, MD, and CR.

| Model | TR ↑ | $R_p$ ↑ | $\sigma_p$ ↓ | SR ↑ | WR ↑ | MD ↑ | CR ↑ |
|---|---|---|---|---|---|---|---|
| DDPG (Lillicrap et al., 2015) | 3.7607 | 0.1122 | 3.0065 | 0.0373 | 58.18 | -17.9332 | 1.0277 |
| PPO (Schulman et al., 2017) | 2.9264 | 0.0808 | 2.3933 | 0.0338 | 56.36 | -15.1331 | 0.9336 |
| A2C (Haarnoja et al., 2018) | 4.7797 | 0.1021 | 1.8610 | 0.0549 | 56.36 | -14.6568 | 1.6274 |
| TD3 (Fujimoto et al., 2018) | 4.7797 | 0.1021 | 1.8610 | 0.0549 | 56.36 | -14.6568 | 1.6274 |
| GPT-3.5-Turbo (OpenAI, 2023) | 2.7849 | 0.0849 | 2.6750 | 0.0317 | 35.71 | -15.7910 | 0.8332 |
| GPT-4.1-mini (Hurst et al., 2024) | 0.7256 | 0.0340 | 2.0602 | 0.0165 | 50.00 | -15.3606 | 0.2153 |
| GPT-5-mini (OpenAI, 2025) | 4.2110 | 0.0810 | 1.2151 | 0.0666 | 58.93 | -7.6517 | 2.6655 |
| LLaMa-3.1 (Dubey et al., 2024) | -2.6938 | -0.0265 | 2.1283 | -0.0125 | 57.14 | -12.2622 | -0.9430 |
| Qwen-3 (Yang et al., 2025) | 2.1798 | 0.0392 | **0.3691** | 0.1062 | 53.57 | -2.2016 | 4.6285 |
| DeepSeek-V3.2 (Liu et al., 2024) | 14.7533 | 0.2660 | 1.9992 | 0.1330 | 60.71 | -12.1658 | 7.0490 |
| Gemini-3-Pro (Gemini Team, 2025) | 4.6656 | 0.1116 | 2.4656 | 0.0453 | 55.36 | -14.3502 | 1.5872 |
| StockSim (Papadakis et al., 2025) *w/* GPT-5-mini | 11.2000 | 0.3121 | 0.3766 | 0.1187 | 40.00 | **8.6800** | 0.3520 |
| ArenaTrader (Ma et al., 2025) *w/* GPT-4.1-mini | 9.2222 | 0.1821 | 2.2197 | 0.0821 | 58.93 | -11.3321 | 4.3002 |
| ArenaTrader (Ma et al., 2025) *w/* GPT-5-mini | 13.8481 | 0.2445 | 1.6031 | 0.1525 | 62.50 | -8.1723 | 9.6977 |
| DeMAgent *w/* GPT-3.5-Turbo | 14.8308 | 0.2790 | 2.5294 | 0.1103 | 58.93 | -13.0209 | 6.6295 |
| DeMAgent *w/* GPT-4.1-mini | 16.4921 | 0.3075 | 2.6606 | 0.1156 | 60.71 | -15.6794 | 6.2988 |
| DeMAgent *w/* GPT-5-mini | 25.2135 | 0.4121 | 1.4159 | **0.2910** | **67.86** | -5.3587 | **32.6687** |
| DeMAgent *w/* LLaMa-3.1 | 6.2413 | 0.1733 | 3.6184 | 0.0479 | 55.36 | -21.3618 | 1.4660 |
| DeMAgent *w/* Qwen-3 | 13.3439 | 0.2388 | 1.7423 | 0.1371 | 50.00 | -9.6094 | 7.8786 |
| DeMAgent *w/* Gemini-3-Pro | 16.2716 | 0.3111 | 2.9253 | 0.1064 | 58.93 | -16.8632 | 5.7566 |
| DeMAgent *w/* DeepSeek-V3.2 | **26.9338** | **0.4540** | 2.3486 | 0.1933 | 57.14 | -10.1719 | 18.9230 |

