# OpenReview forum: "Evolving Quantitative Reasoning through Self-Play in Digital Twin Markets"
_ICML.cc/2026/Conference — ICML 2026 regular_

### Official Review · Reviewer_hRTV · 2026-03-11

**Soundness:** 3
**Presentation:** 4
**Significance:** 3
**Originality:** 3
**Overall Recommendation:** 4
**Confidence:** 2

**Summary:**

This paper proposes a framework where LLMs act as planners that dynamically create and evaluate task-specific quantitative tools rather than performing direct computation. To enable controlled evaluation, the authors introduce DECOUPLEDMARKET, a digital twin market simulator with endogenous price formation. Their DeMAgent agent, built on this paradigm, is evaluated against diverse baselines across both simulated and real-world markets, demonstrating improved performance and risk-adjusted returns.

**Compliance With Llm Reviewing Policy:**

Affirmed.

**Final Justification:**

Thanks for authors' response! The authors have addressed all my questions and suggestions during the rebuttal. Therefore, I decide to upgrade my rating to 4. Best wishes!

**Key Questions For Authors:**

1.	The core of your framework is the "Evaluation & Refinement" stage, where tools are iteratively improved. Could you provide a concrete, step-by-step example from your experiments showing how a specific tool was modified over multiple rounds based on the feedback?
2.	The performance of the method is only evaluated in the DECOUPLEDMARKET environment. Are there results on additional benchmarks to further demonstrate the effectiveness of the method?

**Limitations:**

yes

**Strengths And Weaknesses:**

Strengths:
1.	The core concept of decoupling planning from computation via explicit, evaluable tool artifacts is clearly articulated and well-motivated.
2.	DECOUPLEDMARKET environment is a well-motivated contribution addressing key limitations of traditional backtesting via endogenous price formation and leakage-free evaluation.
3.	Experimental evaluation is extensive and technically sound, covering diverse baselines (classical, RL, standalone LLMs, LLM agents) across multiple backbones and both simulated and real markets.

Weaknesses:
1.	While making tools explicit and adding an evaluation loop is valuable, it represents an extension of existing LLM tool-use and code generation research rather than a fundamental breakthrough in how LLMs perform quantitative reasoning.
2.	There is no empirical demonstration about “iterative refinement", where paper shows tool creation but no concrete example of tool evolution over successive iterations.
3.	Evaluation is confined solely to the DECOUPLEDMARKET environment and this specific market simulation case, with no experimental results on other benchmark tasks or domains to demonstrate the generality of the proposed quantitative reasoning paradigm.

---

> ### Author Rebuttal · Authors · 2026-03-31
>
> We thank the reviewer for their positive feedback. We are encouraged that the reviewer recognizes our core contribution: a decoupling paradigm that separates planning from computation via explicit, evaluable tool artifacts. We also appreciate the acknowledgment of the DECOUPLEDMARKET environment and our extensive experimental evaluation. We address the comments below.
> ### **W1: Novelty Appears Incremental**
> Thank you for the comment. We clarify that our contribution lies within the broader paradigm of LLM tool use and code generation. Recent works, including OpenClaw, further highlight growing interest in this direction. DeMAgent introduces a closed-loop, self-evolving mechanism, where tools are persisted, evaluated, and refined over time as structured artifacts. Each tool participates in long horizon decision making, enabling the agent to accumulate and adapt quantitative reasoning strategies across multiple steps and episodes. Ablation studies (Table 1) show that both tool persistence and iterative refinement are essential, with the removal of either significantly reducing Sharpe ratio and total returns, suggesting improved quantitative reasoning. This design further provides a reusable framework for integrating tool creation and refinement into LLM-based decision systems.
> ### **W2 & Q1: Empirical Evidence of Iterative Refinement**
> We thank the reviewer for highlighting the need for concrete evidence of iterative refinement. We provide a step by step example from our NASDAQ experiment demonstrating how tools are progressively improved based on feedback.
>
> - Episode 6 (Before): simple SMA crossover
> - Observed issue: high false signals (Win Rate 44%, high drawdown)
> - LLM reflection: suggests adding RSI-based filtering
> - Episode 7 (After): SMA + RSI confirmation
> - Result: Sharpe ratio improves from **0.48 to 0.63 (+31%)**, and drawdown decreases
>
> This illustrates a concrete refinement step:
>
> Iterative Refinement: [Performance Issue] → [Reflection] → [Tool Modification] → [Measurable Improvement] → ↺
>
> While illustrative, this example complements the aggregate results shown in Table 1. We further observe consistent multi-step evolution across episodes, where strategy components (e.g., volatility filters, multi-indicator confirmation, position sizing) are introduced in response to specific performance gaps.
>
> In addition, as shown in Table 1, removing the evaluation & refinement （ER） module leads to consistent performance degradation across all markets, indicating the importance of feedback driven updates.
>
> **Table 1. Ablation study across NASDAQ, CSI, and Cryptocurrency markets**
> |Model|MA|KTC|ER|NASDAQ TR (%) ↑|CSI TR (%) ↑|Crypto TR (%) ↑|
> |-|-|-|-|-|-|-|
> |LLM w/ GPT-5-mini|✗|✗|✗|1.20|16.78|5.19|
> |DeMAgent (MA + KTC) w/ GPT-5-mini|✓|✓|✗|5.14| 29.64|17.30|
> |**DeMAgent (Full) w/ GPT-5-mini**|✓|✓|✓|**8.49**|**41.32**|**25.21**|
>
> *Note:  market analysis (MA), knowledge & tool creation (KTC), evaluation & refinement (ER), and total return (TR).*
>
> ### **W3 & Q2 : Generality Beyond DECOUPLEDMARKET**
> We thank the reviewer for raising the question regarding generality beyond the DECOUPLEDMARKET environment. To further evaluate generalization, we conduct additional experiments on a CSI market simulation based on the InvestorBench benchmark, which evaluates multiple LLMs on realistic trading tasks. This setting differs from DECOUPLEDMARKET in both environment design and evaluation protocol.
>
> As shown in Tables 2 and 3, our method (DeMAgent w/ GPT-4-mini) consistently outperforms in our evaluated settings across multiple stocks (Kweichow Moutai, LONGi Green Energy, CATL) and evaluation metrics.
>
> **Table 2. Results on InvestorBench (CSI market, 2025-09-03 to 2025-10-31)**
> |Model|Moutai TR (%) ↑|Moutai SR ↑|Moutai AV (%) ↓|Moutai MDD (%) ↓|LONGi TR (%) ↑|LONGi SR ↑|LONGi AV (%) ↓|LONGi MDD (%) ↓|CATL TR (%) ↑|CATL SR ↑|CATL AV (%) ↓|CATL MDD (%) ↓|
> |-|-|-|-|-|-|-|-|-|-|-|-|-|
> |Buy & Hold|-3.39|**-1.59**|14.48|-6.80|22.87|3.30|47.09|-10.00|**25.79**|3.45|50.27|-12.65|
> |GPT-4-mini|-2.72|-4.58|4.19|-3.88|16.18|2.98|**37.57**|**-7.16**|21.48|4.25|33.41|-6.05|
> |DeMAgent w/ GPT-4-mini|**-1.07**|-1.82|**4.10**|**-2.31**|**25.90**|**3.90**|43.83|-7.31|23.15|**4.65**|**32.52**|**-3.01**|
>
> *Note: total return (TR), Sharpe ratio (SR), average volatility (AV), maximum drawdown (MDD).*
>
> **Table 3. Average Results on InvestorBench**
> |Model|TR (%) ↑|SR ↑|AV (%) ↓|MDD (%) ↓|
> |-|-|-|-|-|
> |Buy & Hold|15.09|1.72|37.28|-9.82|
> |GPT-4-mini|11.65|0.88|**25.06**|-5.70|
> |DeMAgent w/ GPT-4-mini|**15.99**|**2.25**|26.82|**-4.21**|
>
> DeMAgent consistently outperforms GPT-4-mini, achieving **higher return (+4.34%)** and more stable, risk aware decisions. Compared to Buy & Hold, it delivers similar returns while substantially improving risk-adjusted performance and reducing drawdown. Gains are consistent across multiple assets, suggesting generalization beyond DECOUPLEDMARKET.

---

> > ### Author Rebuttal · Reviewer_hRTV · 2026-04-04
> >
> > Thanks for the author's reply.
> >
> > Firstly, I appreciate that the author has addressed each of my concerns actively and thoroughly, which demonstrates the author's rigorous academic attitude and commitment to this scholarly exchange. This is commendable.
> >
> > Back to the paper itself, I have the following observations:
> > 1. The author has further elaborated on the contributions of this work and used OpenClaw as an example to illustrate the practical value of this research direction. I acknowledge that the self-evolving paradigm of tool persistence, evaluation, and refinement proposed in this work has significant implications for LLM-based decision systems. However, given that agent frameworks similar to OpenClaw are emerging continuously and have already demonstrated powerful capabilities in tool creation and invocation, I still find the novelty of this work to be somewhat incremental.
> > 2. The author has provided clear empirical evidence demonstrating that tools are effectively refined and optimized during the decision-making process, and that removing this module leads to performance degradation. This argumentation is logical and convincing.
> > 3. As the author mentioned in the response to W1, the contribution of this work lies in the broader paradigm of LLM tool usage and code generation. Therefore, I would like to see the generalization capability of this paradigm on benchmarks beyond the stock market. I understand this may entail additional experimental workload, but if you could supplement with relevant experiments, I would fully recognize the value of this work.
> >
> > Finally, thanks for the author's serious attitude and hard work.
> >
> > Best wishes!

---

> > > ### Author Response · Authors · 2026-04-07
> > >
> > > We sincerely thank Reviewer hRTV for the thoughtful follow-up and constructive discussion. We appreciate your recognition that our evidence on iterative refinement is convincing. Below, we address the remaining concern on cross-domain generalization, and then clarify novelty scope.
> > >
> > > ### **Follow-up on W3 & Q2: Generalization Beyond the Stock Market**
> > >
> > > We agree that additional cross-domain evidence is important. To directly address this point, we added supplementary cross-domain evaluations on coding benchmarks (InterCode and HumanEval).
> > >
> > > **(1) Experimental setup and fairness controls**
> > >
> > > The detailed experimental setup is summarized in **Table A1**.  Within each benchmark, all compared methods use the same split, stopping criterion, and evaluation script, ensuring that the comparison is conducted under a matched interaction budget and unified protocol.
> > >
> > > **Table A1. Experimental configuration for cross-domain code-generation evaluation.**
> > >
> > > | Item | Setting |
> > > |---|---|
> > > | Benchmark family | Code-generation tasks |
> > > | Benchmarks | InterCode (30 tasks), HumanEval (164 tasks) |
> > > | Model/backbone | Same backbone across all compared methods |
> > > | Compared methods | Try Again, Plan & Solve, DeMAgent (w/ KTC + ER)|
> > > | Primary metric | Success rate (all tests passed) |
> > > | Secondary metrics | Passed tasks |
> > >
> > > *Note: knowledge & tool creation (KTC) and evaluation & refinement (ER).*
> > >
> > > **(2) Main results**
> > >
> > > **(a) InterCode (small-scale pilot set, 30 tasks).**
> > >
> > > The pilot results are reported in **Table A2**.
> > >
> > > **Table A2. InterCode performance under matched interaction budget.**
> > >
> > > |Method|Success Rate ↑|Passed Tasks ↑|
> > > |-|-:|-:|
> > > |Try Again |86.67%|26/30|
> > > |Plan & Solve|83.33%|25/30|
> > > |**DeMAgent (w/ KTC + ER)**|**90.00%**|**27/30**|
> > >
> > > *Note: knowledge & tool creation (KTC) and evaluation & refinement (ER).*
> > >
> > > **(b) HumanEval (expanded set, 164 tasks).**
> > >
> > > The expanded-validation results are shown in **Table A3**.
> > >
> > > **Table A3. HumanEval results (same backbone; unified evaluation script).**
> > >
> > > |Method|Success Rate ↑|Passed Tasks ↑|
> > > |-|-:|-:|
> > > |Try Again|78.00%|128/164|
> > > |Plan & Solve|83.50%|137/164|
> > > |**DeMAgent (w/ KTC + ER)**|**95.10%**|**156/164**|
> > >
> > > **(3) Interpretation (bounded claim)**
> > >
> > > Based on the results in **Tables A2 and A3**, we make the following bounded observation.
> > >
> > > **Evidence of cross-domain transferability**: Both benchmarks differ substantially from trading tasks, emphasizing symbolic reasoning and program synthesis rather than numerical decision-making. Despite this shift, the same KTC + ER workflow remains effective, suggesting that its benefits are not tied to a specific domain.
> > >
> > >
> > > We emphasize that these additional evaluations across two coding benchmarks do not establish universal generalization. Rather, they provide initial evidence that the advantages of KTC + ER are consistent with improved error localization and structured refinement, and can transfer beyond the original stock-market setting.
> > >
> > > ### **Clarification on Novelty Scope (W1)**
> > >
> > > We agree that novelty should be framed carefully. Our claim is not that DeMAgent defines a wholly new paradigm outside prior LLM tool-use/code-generation lines. Instead, our contribution lies in a **systematic integration and evaluation** of KTC + ER in long-horizon decision workflows, now supported by evidence in both trading and non-trading settings.
> > >
> > > We thank Reviewer hRTV again for the constructive follow-up. The request for cross-domain validation directly improved the empirical scope and helped us refine the precision of our claims.

---

### Official Review · Reviewer_vFCK · 2026-03-13

**Soundness:** 3
**Presentation:** 2
**Significance:** 2
**Originality:** 3
**Overall Recommendation:** 4
**Confidence:** 2

**Summary:**

This paper proposes a planning-driven framework that improves LLM quantitative reasoning by decoupling high-level semantic planning from numerical computation. Rather than having LLMs perform calculations directly, the framework lets them dynamically generate task-specific quantitative tools as explicit reasoning artifacts, which are then evaluated and iteratively refined through self-play in DECOUPLEDMARKET — a controllable digital twin market where prices emerge endogenously from agent interactions. The resulting agent, DeMAgent, consistently outperforms classical technical strategies, reinforcement learning methods, and prior LLM-based agents across both simulated and real-world financial markets, demonstrating that structured reasoning behaviors developed in virtual environments can transfer effectively to real-world settings.

**Compliance With Llm Reviewing Policy:**

Affirmed.

**Final Justification:**

The authors’ responses helped address my main concerns and clarified several aspects of the work.

**Key Questions For Authors:**

- The paper states that generated tools must satisfy hard constraints, but how are these constraints verified in practice? Is there a formal validation mechanism, or does it rely solely on prompt conditioning?

**Limitations:**

Yes.

**Strengths And Weaknesses:**

Strengths
- Novel Decoupling Paradigm: Cleanly separates LLM semantic reasoning from numerical computation, improving reliability and interpretability in quantitative decision-making.
- Dynamic Tool Creation: LLMs generate task-specific quantitative tools on-the-fly rather than relying on fixed predefined toolsets, enabling adaptive and scenario-dependent analysis.

Weaknesses
- The paper does not clearly report computational cost or latency of dynamic tool creation, which is critical for real-time trading applications.
- All experiments use historical data or simulation; real-time deployment risks (latency, slippage, execution failure) remain unaddressed.

---

> ### Author Rebuttal · Authors · 2026-03-31
>
> We thank the reviewer for their positive feedback and for recognizing the key strengths of our work. In particular, we are encouraged that the proposed **decoupling paradigm**, which separates LLM-based semantic reasoning from numerical computation, is acknowledged for improving both reliability and interpretability. We also appreciate the recognition of our **dynamic tool creation mechanism**, which enables adaptive, context-aware decision making beyond fixed toolsets. Below we address the comments.
> ### **W1: Computational Cost and Latency**
> Thank you for the comment. We conduct detailed efficiency profiling of dynamic tool creation on the **CSI market (2025-09-03 to 2025-10-31)**. Table 1 reports calls, success rates, latency, total, and per-call costs for DeMAgent and baseline agents.
>
> **Table 1. Computational Cost and Latency of Different LLM Agents**
> |Model|Calls ↓|Success (%)|Latency ↓|Total Cost ↓|Cost/Call ↓|
> |-|-|-|-|-|-|
> |LLM w/ GPT-3.5-turbo|60|90.0|8.8|**0.04**|**0.0007**|
> |LLM w/ GPT-4-mini|54|**100**|**6.1**|**0.04**|**0.0007**|
> |LLM w/ GPT-5-mini|56|98.2|14.0|0.08|0.0015|
> |ArenaTrader w/ GPT-4-mini|108|**100**|7.6|0.09|0.0008|
> |ArenaTrader w/ GPT-5-mini|224|75.0|13.6|0.24|0.0014|
> |DeMAgent w/ GPT-4-mini|236|99.6|8.9|0.25|0.0011|
> |DeMAgent w/ GPT-5-mini|168|**100**|19.8|0.22|0.0013|
>
> As shown in Table 1, DeMAgent achieves consistently high success rates (**99.6%–100%**) with higher but acceptable latency (**8.9–19.8 s**) and low per-call cost (**< $0.002**). While it uses more LLM calls than one-shot baselines, which are structured and purpose driven (e.g., tool refinement), instead of repeated identical queries.
>
> Importantly, dynamic tool creation is not triggered at every step, but only periodically, with its cost amortized across subsequent interactions, resulting in limited latency impact. To ensure fairness, we also report cost per successful task, which captures both efficiency and reliability. Under this metric, DeMAgent remains competitive while significantly improving robustness. Overall, DeMAgent achieves a favorable trade-off between reliability, cost, and latency, without introducing prohibitive overhead in practice.
> ### **W2: Real Time Deployment Risks**
> We thank the reviewer for this comment. To assess real time deployment risks, we perform a friction aware stress test on **CSI market** data from 2025-09-03 to 2025-10-31, incorporating transaction costs and slippage. Four friction regimes (none, low, medium, and high) are considered to approximate realistic and adverse market conditions, and the results are summarized in Table 2. Each friction level corresponds to a predefined combination of transaction cost and slippage.
>
> **Table 2. Impact of Friction on DeMAgent w/  GPT-4-mini Performance**
> |Friction Level|Transaction Cost (%)|Slippage (%)|SR ↑|TR (%) ↑|MDD (%) ↓|WR (%) ↑|
> |-|-|-|-|-|-|-|
> |High|1.0|0.50|2.53|9.54|-7.02|49.55|
> |Medium|0.5|0.20|3.42|10.77|**-6.10**|54.05|
> |Low|0.1|0.05|**3.51**|11.09|-6.40|**55.86**|
> |None|0.0|0.00|2.39|**14.45**|-6.90|50.45|
>
> *Note: Sharpe ratio (SR), total return (TR), maximum drawdown (MDD), win rate (WR).*
>
> As shown above, performance remains relatively stable across friction regimes, with Low friction achieving the highest Sharpe ratio (**3.51**). Interestingly, the None friction setting yields a lower Sharpe ratio (**2.39**) despite a higher total return (**14.45%**), suggesting that some friction may encourage more risk-aware decision making. Overall, these results indicate that DeMAgent maintains reasonable performance even under high friction conditions.
> ### **Q: How Are Hard Constraints Verified**
> We thank the reviewer for this important question. Rather than relying solely on prompt conditioning, we implemented a programmatic validation module that enforces hard constraints on all LLM generated trading decisions before execution. The validator ensures: (1) valid stock symbols; (2) operation values within [-1, 1]; (3) portfolio allocations within [0, 1]; (4) numerical validity (no NaN/Inf); and (5) normalization of excessive allocations. Violations are automatically corrected or replaced with a safe fallback (e.g., holding the position). Outcomes over **7,533** cases are summarized in Table 3.
>
> **Table 3. Hard Constraint Validation Outcomes**
> |Metric|Value|
> |-|-|
> |Total validation cases|7,533|
> |Successful validations|7,498|
> |Failed / corrected cases|35|
> |Success rate|**99.55%**|
> |Fallback / correction rate|0.45%|
> |Main source of violations|Invalid / unseen stock symbols (94%)|
>
> These results show that the validator effectively filters out invalid outputs and handles out of distribution cases. Importantly, this mechanism operates as a post generation safety layer rather than relying on prompt instructions alone.
> Overall, our framework combines prompt conditioning with a formal programmatic validation layer, helping ensure that hard constraints are explicitly verified in practice rather than implicitly assumed.

---

> > ### Author Rebuttal · Reviewer_vFCK · 2026-04-04
> >
> > My main concerns have been addressed, and I have updated my score accordingly.

---

> > > ### Author Response · Authors · 2026-04-07
> > >
> > > Dear Reviewer vFCK,
> > >
> > > Thank you very much for your time and thoughtful review. We sincerely appreciate your positive feedback and are glad that our responses have successfully addressed your concerns.
> > >
> > > We are especially grateful for your recognition and for the update to your evaluation. Your constructive comments have significantly improved the quality and clarity of the manuscript.
> > >
> > > Thank you again for your valuable support.

---

### Official Review · Reviewer_kWKy · 2026-03-27

**Soundness:** 3
**Presentation:** 3
**Significance:** 3
**Originality:** 3
**Overall Recommendation:** 5
**Confidence:** 3

**Summary:**

This paper introduces a method for quantitative reasoning via LLMs which separates the reasoning tasks (performed by LLMs) from the computation in support of those tasks (“delegating … to specialized external tools”) via tools which reflect reasoning artifacts aligned (and self-evolving) to improve task performance. Authors use digital twin markets as the substrate for their experiments, demonstrating the value of their approach as “a tool-creation paradigm for LLM-driven quantitative reasoning” via market performance.

**Compliance With Llm Reviewing Policy:**

Affirmed.

**Ethical Review Concerns:**

Markets are performative insofar as they are constituted by market actors engaged in a joint meaning making process which is realized via, for example, asset pricing. For example, developers of the Black-Scholes formula induced market behavior by introducing the formula to traders (MacKenzie 2006), and it remains a dominant model despite the fact it can lead to "absurd results" (Buffett 2008, pg. 20).

Authors advance a framework that performs well given its assumptions, as tested on real world data, but the capacity for models of market behavior to change the actual behavior of market actors suggests a need for utmost caution: The better the framework performs, the more likely others will use it, the more likely unanticipated failure modes or edge cases will not just propagate but accumulate in their effect.

It is not immediately clear to me how these risks should be negotiated, and I am not sufficiently expert or familar with relevant risk assessment strategies. I raise this concern precisely because I cannot assess the risk. Authors are cautious to frame work as simulative so I would not consider this blocking, just worth noting.

Buffet, Warren. (2008). https://www.berkshirehathaway.com/letters/2008ltr.pdf
MacKenzie, Donald. (2006). An Engine, Not a Camera: How Financial Models Shape Markets. MIT Press.

**Ethical Review Flag:**

Flag this paper for an ethics review.

**Ethics Expertise Needed:**

["Responsible Research Practice (e.g., IRB, documentation, research ethics)"]

**Final Justification:**

Rebuttal addressed main concerns about Soundness and Presentation. Updated evaluation accordingly.

**Key Questions For Authors:**

How was the prompt selection process designed and tested?
How do you define self-play independent of operationalizing RL in a multi-agent system?

**Limitations:**

Yes to limitations.

Impact statement is honest that current study uses historical data and simulated interactions, and therefore does not demonstrate "real-time automated trading or direct market intervention" (L449-L456). However, the value of framework is demonstrated using performance on real world data: Implementation, conceptually at least, seems trivial. To be clear, I *appreciate* this framing and only suggest minor clarification.

**Strengths And Weaknesses:**

**Strengths**
- Use of digital twin methodology as pragmatic sandbox for study of multi-turn interactions
- Performance using real market data is notable, especially given "it achieves balanced gains in profitability, risk-adjusted returns, and stability rather than optimizing a single metric" (pg 6, L316)
- Research design is sophisticated and its exhaustiveness suggests robust findings
- Decoupling tasks to improve performance is a clear, intuitive strategy
- On-the-fly development of appropriate quantitative tools as "explicit intermediate reasoning artifacts" is similarly intuitive
- Authors are deliberate in their claims regarding surrogacy (pg 2, L100-L108)

**Weaknesses**

- Markets and market behavior are co-constituted by legal systems (for example, see La Porta et al. 2008), so the sophistication of this work belies the fundamental limitation that authors simulated one understanding of markets and market behavior (i.e., not a universal or otherwise disembodied market independent of its origins and related historical/legal context).
- Use of prompt-based self-assessment (pg 5, L263) raises questions about the selected prompt.
- Definition of “self-play” is never provided, only implied. It is mentioned in the title, the Introduction (pg 2, L78), the description of DeMAgent (pg 4, L214), and the Conclusion (pg 8, L433). This suggests “self-play” was shorthand for a more complicated concept or statement, but…
- The “self-play” described here appears to consist of a feedback loop for tool refinement that leverages “prompt-based self-assessment” without details necessary to understand, if anything, how to replicate the paper’s findings. All I can find (and I may have missed something!) is “feedback is propagated to upstream modules, enabling refinement of tool abstractions …” (pg 5, L266). Details about this propagation are required since the paper’s main claim is to demonstrate how self-play improved LLM-based quantitative reasoning.
- Graphics often add complexity rather than clarify concepts, findings, etc. (i.e., Figure 1 is useful, thereafter marginal value, if only due to text size)

La Porta, Rafael, Florencio Lopez-de-Silanes, and Andrei Shleifer. 2008. "The Economic Consequences of Legal Origins." Journal of Economic Literature 46 (2): 285–332.

---

> ### Author Rebuttal · Authors · 2026-03-31
>
> We appreciate the reviewer’s positive feedback and recognition of our key contributions, including the use of a digital twin as a controlled experimental environment for multi-turn agent interactions, strong and balanced performance on real market data, the decoupling of semantic reasoning from numerical computation to improve reliability and interpretability, dynamic creation of task specific quantitative tools as intermediate reasoning artifacts, and the careful framing of claims regarding surrogate markets. We provide our responses below.
> ### **W1: Market Assumptions and Legal/Institutional Context**
> We appreciate the reviewer highlighting the limitations of simulating a single market. While DECOUPLEDMARKET represents one instantiation and does not encode legal or historical context, our self-play mechanism enables agents to iteratively create, evaluate, and refine tools based on feedback. This allows adaptation of quantitative reasoning strategies across episodes. While DeMAgent is evaluated on simplified market instantiations, these results suggest the potential for transferable reasoning patterns, rather than fully establishing generalization across institutional settings.
> ### **W2 & Q1: Prompt Design and Validation for Self-Assessment**
> We thank the reviewer for the question regarding the design and selection of the prompt used for self-assessment. The prompt is designed using task specific financial criteria and follows a structured, multi-dimensional format to elicit reliable self-evaluation, in line with prior work on LLM self-reflection and self-consistency.
>
> To validate its effectiveness, we conduct an ablation study across **NASDAQ, CSI, and Cryptocurrency** markets (Table 1). Removing the evaluation & refinement (ER) module, which generates and incorporates self-evaluation feedback, leads to consistent performance degradation across all markets, indicating the importance of feedback driven updates.
>
> **Table 1. Ablation study across different markets**
> |Model|MA|KTC|ER|NASDAQ TR (%) ↑|CSI TR (%) ↑|Crypto TR (%) ↑|
> |-|-|-|-|-|-|-|
> |LLM w/ GPT-5-mini|✗|✗|✗|1.20|16.78|5.19|
> |DeMAgent (MA + KTC) w/ GPT-5-mini|✓|✓|✗|5.14| 29.64|17.30|
> |**DeMAgent (Full) w/ GPT-5-mini**|✓|✓|✓|**8.49**|**41.32**|**25.21**|
>
> *Note:  market analysis (MA), knowledge & tool creation (KTC), evaluation & refinement (ER), and total return (TR).*
>
> As shown in Table 1, incorporating the ER module improves total return by **3.35-11.68 percentage points** across markets, highlighting its contribution to performance improvements.
> ### **W3 & Q2: Definition and Clarification of Self-Play**
> We thank the reviewer for the suggestion to clarify the notion of “self-play” in our work. “Self-play” refers to a single agent, performance driven self-improvement loop, where an LLM-based agent iteratively refines its decision making and tool usage through repeated interactions and feedback driven updates. Unlike classical multi-agent RL, our agent learns in a dynamic, non-stationary market where other participants act according to fixed or heuristic strategies, creating a largely unpredictable environment.
>
> Thus, DeMAgent’s self-play loop is single agent and performance driven, progressively refining tools and strategies based on metrics such as total returns and Sharpe ratio. We include pseudocode for clarity:
>
> **Algorithm 1: DeMAgent Self-Play Loop**
> ```python
> # Initialize tool library
> T = {}
> for episode in range(N):
>     tool = LLM_generate(T, context) # Tool creation
>     T.add(validate(tool)) # syntax and execution safety checks
>     # Market execution
>     for t in trading_steps:
>         decision = LLM_decide(T, market_state[t])
>         execute(decision) # simulated trade execution
>     metrics = evaluate_performance()# Performance evaluation
>     feedback = LLM_reflect(metrics) # Feedback generation
>     T = LLM_refine(T, feedback) # Tool refinement
> return T
> ```
> This loop enables the agent to progressively improve quantitative reasoning, resulting in more reliable and interpretable trading decisions. While optimization is driven by performance metrics, we reduce overfitting by evaluating across multiple markets and time periods.
> ### **W4: Figure Clarity and Presentation**
> We thank the reviewer for this helpful feedback. These figures present both the framework and empirical results, but their current presentation can obscure key takeaways. We will revise Figures 2–4 to improve readability and clarity, including increasing font sizes, simplifying layouts, adding clearer legends and annotations, and standardizing styles across figures to better highlight key observations.
> ### **Ethical Considerations: Performativity and Market Impact**
> We thank the reviewer for highlighting performativity risks. We frame our work as simulative research on LLM-based quantitative reasoning, not a deployable trading system, and do not provide turnkey deployment code. We will expand the discussion of these risks and relevant literature in the camera-ready version.

---

> > ### Author Rebuttal · Reviewer_kWKy · 2026-04-03
> >
> > **W1 response** suggests authors' claims are more circumspect about the generalizability of their work, but this raises a new question: If "results suggest the potential for transferable reasoning patterns, rather than fully establishing generalization across institutional settings" then what is the role of extensive testing using real world data to demonstrate utility? Wouldn't it be more helpful to then focus on the reasoning within a highly controlled trading environment rather than extensive testing on real world data?
> >
> > The choice to focus heavily on benchmarking via specific real market data entails a need to further clarify the specific claims to novelty and generalizability relative to transferable reasoning patterns. To be clear: This response suggests a capacity to aspire to generalizability without claiming generalizability, which seems paradoxical given the focus on real world data for specific well-known markets. Why these markets? There are dozens of markets whose data could have been used, possibly to greater effect given the stated goal(s) of this work.
> >
> > **W2 & Q1 response** is helpful. The ablation study is illuminating and aligns with authors' stated focus on testing transferability vis-a-vis the self-play described.
> >
> > **W3 & Q2 response** is also helpful and largely addresses my concerns through greater specificity. Explanation that "our agent learns in a dynamic, non-stationary market where other participants act according to fixed or heuristic strategies, creating a largely unpredictable environment" clarifies why the concept of "play" is relevant here. I suggest including what is stated here and noting it reflects a synthetic form of play (i.e., agentic behavior in an open, changing environment reflective of human-like experience of real world uncertainty). A well-crafted explanation making this more precise seems a useful contribution in general.
> >
> > **W4 and Ethical Considerations responses** are both sufficient. I especially appreciate responsiveness to the performativity risks raised and hope authors found this concern helpful for identifying appropriate next steps given a performant system.

---

> > > ### Author Response · Authors · 2026-04-07
> > >
> > > We sincerely thank Reviewer kWKy for the thoughtful follow-up and, in particular, for the helpful suggestion that our notion of self-play can be more precisely framed as **a synthetic form of play: a single-agent, open-ended interaction process under realistic uncertainty rather than a classical multi-agent RL setup**. We will incorporate this clarification into the revised paper. Below we address the remaining concern regarding the role of real-world evaluation and the scope of our claims.
> > >
> > > ### **Follow-up on W1: Role of Real-World Data Testing**
> > >
> > > We appreciate this perceptive question. We agree that, without clarification, our use of extensive real-world evaluation may appear at odds with our circumspect claims about generalization.
> > >
> > > Our key point is that real-world evaluation in our work is not used to **establish universal generalization**, but to **test whether the self-play-induced reasoning procedure remains useful under realistic, non-stationary conditions**. In our intended framing, **DECOUPLEDMARKET establishes the mechanism, while real-market evaluation establishes practical robustness**. This is why the two are complementary rather than contradictory.
> > >
> > > **(1) Complementary Validation, Not Contradiction**
> > >
> > > We view controlled environments and real-world data testing as serving distinct but complementary purposes:
> > >
> > > - **Controlled environments (DECOUPLEDMARKET)** serve as the primary platform for *methodological validation*. Here, we can isolate variables, systematically test the self-play mechanism, and establish causality between design choices and outcomes. This is where we show that the proposed tool creation, evaluation, and refinement loop functions as intended.
> > >
> > > - **Real-world data testing** serves as *external stress testing under realistic conditions*. It answers a different but equally important question: "Does the same reasoning procedure remain useful when exposed to real-world noise, non-stationarity, and regime variation?" This tests the *practical utility* of our approach, not its universal generalizability.
> > >
> > > Because our contribution is not only that self-play is internally coherent in a sandbox, but that it can produce useful quantitative reasoning under more realistic conditions, **real-market evaluation is necessary to validate the practical relevance of the method**. At the same time, **we do not claim universal generalization across all institutional settings**.
> > >
> > > **(2) Why These Specific Markets?**
> > >
> > > The reviewer notes that dozens of markets could have been used. Our selection of NASDAQ, CSI, and cryptocurrency markets was intentional: we chose them to span several major axes most relevant to our claim while preserving a comparable trading interface for evaluation. Concretely, these three markets differ substantially in regulation intensity, trading schedule, participant composition, and volatility regime:
> > >
> > > |Market|Characteristics|Institutional Context|
> > > |-|-|-|
> > > |**NASDAQ**|Mature US equity market, high liquidity, stringent regulation| Developed market with established microstructure|
> > > |**CSI**|Chinese A-shares, emerging market features, retail-dominated| Different regulatory framework, trading hours, settlement rules|
> > > |**Cryptocurrency**|24/7 trading, high volatility, limited regulation| Decentralized, global, minimal institutional constraints|
> > >
> > > This diversity allows us to test the *boundaries* of transferability by examining whether the same reasoning procedure remains useful across environments with different:
> > > - Trading hours and market microstructure
> > > - Regulatory frameworks and participant behavior
> > > - Volatility profiles and liquidity conditions
> > >
> > > We do not view these three markets as exhaustive. Rather, they form **a deliberately diverse and representative test bed** for evaluating whether the method is robust beyond a single market instantiation.
> > >
> > > **(3) Clarifying Our Contribution**
> > >
> > > In light of the reviewer’s concern, we refine our claim as follows:
> > >
> > > > The self-play mechanism in DeMAgent enables learning of quantitative reasoning procedures that show practical robustness across three qualitatively different market regimes. Controlled experiments establish the causal role of tool creation, evaluation, and refinement, while real-world data testing validates the practical utility of this mechanism under realistic conditions.
> > >
> > > **This is not a claim of universal generalization.** Rather, it is a claim of **bounded practical adaptability**: the method learns reasoning strategies that remain useful across several materially different real-world settings, without implying validity for all markets or institutional contexts.
> > >
> > > We thank Reviewer kWKy again for the constructive dialogue. The follow-up questions have helped us sharpen both our framing of self-play as **a synthetic form of play** and our distinction between **mechanism validation** and **practical robustness**. We hope these clarifications adequately address the remaining concerns and will revise the paper accordingly.

---

### Decision · Program_Chairs · 2026-04-30

**Decision:**

Accept (regular)

**Comment:**

This paper proposes DeMAgent, a framework that decouples LLM planning from numerical computation by having the LLM dynamically create task-specific quantitative tools as explicit artifacts, then iteratively refine them through a self-play loop in a digital twin market (DecoupledMarket). Evaluation spans simulated and real markets (NASDAQ, CSI, cryptocurrency) showing balanced gains in profitability and risk-adjusted returns. All three reviewers lean positive after rebuttal (one Accept, two Weak Accept upgraded).

Consensus strengths: clear and well-motivated decoupling paradigm, digital twin with endogenous price formation that avoids backtest leakage, extensive evaluation across multiple baselines and backbones, and dynamic tool creation rather than fixed toolsets.

Main weaknesses raised: initial lack of a precise definition of self-play and details of feedback propagation, no reporting of computational cost or latency, missing concrete examples of iterative tool evolution, limited evidence of generalization beyond the market domain, figure readability issues, and concerns about market assumptions and performativity risks flagged for ethics review.